# National identity predicts public health support during a global pandemic

Changing collective behaviour and supporting non-pharmaceutical interventions is an important component in mitigating virus transmission during a pandemic. In a large international collaboration (Study 1, $N = 49,968$ across 67 countries), we investigated self-reported factors associated with public health behaviours (e.g., spatial distancing and stricter hygiene) and endorsed public policy interventions (e.g., closing bars and restaurants) during the early stage of the COVID-19 pandemic (April-May 2020). Respondents who reported identifying more strongly with their nation consistently reported greater engagement in public health behaviours and support for public health policies. Results were similar for representative and non-representative national samples. Study 2 ($N = 42$ countries) conceptually replicated the central finding using aggregate indices of national identity (obtained using the World Values Survey) and a measure of actual behaviour change during the pandemic (obtained from Google mobility reports). Higher levels of national identification prior to the pandemic predicted lower mobility during the early stage of the pandemic ($r = -0.40$). We discuss the potential implications of links between national identity, leadership, and public health for managing COVID-19 and future pandemics.

As of October 2021, more than 235 million people worldwide have been infected by the new coronavirus and nearly 5 million have died[1], making the COVID-19 pandemic one of the greatest health crises of the past century. Until a vaccine or effective medical treatment is widely administered, the public response to the pandemic is largely limited to non-pharmaceutical interventions, including policy-making and collective behaviour change[2]. To reduce virus transmission, it is crucial that people engage in public health behaviour (e.g., maintain spatial distance and improve physical hygiene) and support COVID-19 protective policies (e.g., limiting travel and closing bars and restaurants). And even after effective vaccines are administered, it is critical to convince people to take them. This is why the Director of the World Health Organization declared: *"That's why behavioural science is so important – it helps us to understand how people make decisions, so we can support them to make the best decisions for their health"*[3].

In the current investigation, we respond to this call for behavioural science on the pandemic. Specifically, we present the results from two large-scale global studies across 67 (Study 1) and 42 (Study 2) countries, testing key predictors of public health support during COVID-19. Focusing on the potential role of national identity, we examine the role of key *social* motives in collective behaviour during the pandemic. This research may help scholars, health organizations, and political leaders identify important factors and design more effective behavioural interventions to increase compliance with actions such as maintaining spatial distance and restricting travel during a pandemic.

During a global pandemic, leaders and public health officials need to inform and mobilize the public to avoid behaviours no longer considered socially responsible. However, recent evidence suggests this type of leadership requires cultivating a shared sense of solidarity to increase compliance with recommended health behaviours[4–6]. Solidarity with other members of one's group is a component of ingroup identification[7], that is, the personal significance that being part of a group (e.g., nation) holds for an individual[7–10]. Identifying with a group is associated with mutual cooperation and adherence to its norms[11–13], motivation to help other members of their group[14,15], and a willingness to engage in collectively-oriented actions aimed at improving the group's welfare[10,16–18]. Here we test the role of identification with one's national group in promoting public health in the COVID-19 pandemic (see ref. [19]).

National identity plays an important role in motivating civic involvement[20] and costly behaviours that benefit other members of their national community[21]. Accordingly, a strong sense of shared national identity might help collective efforts to combat the pandemic within a country (e.g., ref. [22]). Moreover, border closures, travel bans, and national task forces have likely made national identities even more salient during the pandemic[23]. The existence and activation of strong collective identities can allow political leaders to mobilize large populations to adhere to emergency public health measures. For instance, political leaders and public health officials often foster a sense that "we are in this together" and that we can manage the crisis through collective action[18,24]. This might be particularly important for counteracting polarization within countries, which can reduce health behaviour and increase the risk for infections and mortality[19,25,26].

The goal of the current paper is to examine whether national identification (NI) is associated with global adherence to the public health measures during a pandemic[27–29]. Specifically, we examined the associations between the strength of identification with one's nation and whether people adopted public health behaviours (e.g., limiting travel, spatial distancing, hand washing) and endorsed public policy interventions (e.g., closing bars and restaurants). Extensive evidence suggests these actions could substantially reduce the number of COVID-19 infections[2,30–32]. Our primary hypothesis is that stronger NI will be associated with greater support for and compliance with public health measures.

National identity is distinct from beliefs about national superiority or collective narcissism (e.g., refs. [33–35]). NN is a form of social identity that involves the belief that one's group (i.e., nation) is exceptional but unappreciated by others[36]. NI tends to correlate positively with NN because they both involve a positive evaluation of one's nation. However, they are linked to very different outcomes. For example, outgroup prejudice is *negatively* associated with NI but *positively* with NN[37].

People high in collective narcissism are especially concerned with how their group reflects on them[38]. For instance, NN is associated with a greater preoccupation with maintaining a positive image of the nation than with the well-being of fellow citizens[39,40]. Thus, in a crisis, national narcissists may prefer to invest in short-term image enhancement rather than in the sorts of long-term solutions that are necessary to sustain public health during a long pandemic (see also ref. [41]). They may then be less inclined to engage in behaviours to prevent the spread of COVID-19 (see ref. [42])--or even acknowledge the risks associated with the pandemic in their home country (e.g., ref. [43]). Therefore, in identifying associations with compliance with public health measures, we sought to distinguish NI from NN.

In addition, there is some evidence that right-wing political ideology (PI) is associated both with national identity (e.g., ref. [44]) and NN (e.g., refs. [39,45,46]). Moreover, supporters of right-wing political parties have tended to downplay risks associated with COVID-19 (e.g., refs. [47–49]) and were less likely to comply with preventative measures compared to left-leaning or liberal individuals[26,48,50]. Therefore, we examined whether NI and narcissism were distinct from PI in explaining public health support.

## Results

The COVID-19 pandemic is a truly *global* crisis with over 200 countries reporting infections. To understand the variables that account for public health support around the globe, we launched a collaborative, international project in April 2020 collecting large-scale data from as many nations as possible. In Study 1, we collected a large sample consisting of citizens from 67 countries. We analyzed a sample of 49,968 participants (see Fig. 1). See "Methods" for details about the sample (all reported materials and data are available at: https://osf.io/y7ckt/).

We analyzed these data using multi-level models in which persons were treated as nested within countries[51]. We also included a measurement level to control for individual differences in how consistently people responded to items that were meant to measure the same construct. Our analyses estimated relationships at the individual level while controlling for country-level differences. For example, did people who had a stronger NI endorse public health measures such as spatial distancing (e.g., reducing physical contact) more strongly than people with a weaker NI? A set of regression coefficients was estimated for each country, and the means of these coefficients were tested for statistical significance. Moreover, the standard errors of these coefficient incorporated "Bayesian shrinkage" meaning that less reliable observations (countries and individuals) influenced parameter estimates less than more reliable observations.

We also adjusted for the COVID-19 infection and mortality rates within each country to ensure that public health support was not merely a function of local risks. Due to the large sample size in Study 1, we focused our interpretations on the person-level findings that were statistically significant at the $p < 0.001$ level. (The results with the Human Development Index (HDI) are available in the Supplementary Information).

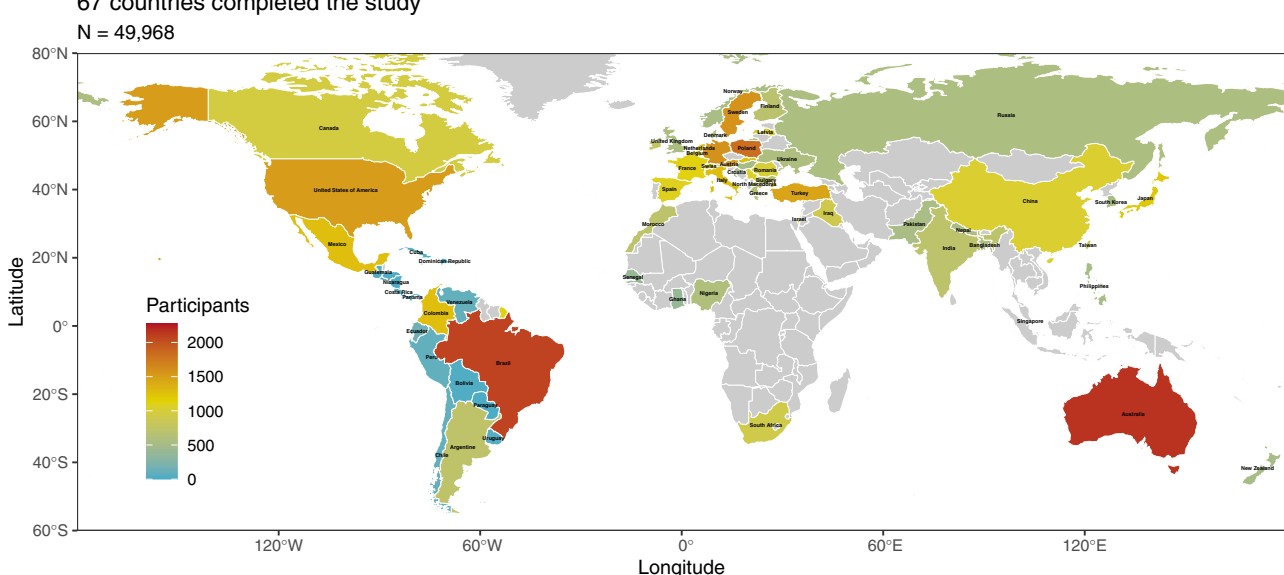

**Fig. 1 Map of the 67 participating countries and territories with total sample size scaled to colour (we did not obtain samples from countries in grey).**
All the worldmaps were produced using R packages. The map is from the package 'rworldmap' and is licensed-free from South, A. (2011). rworldmap: A New R package for Mapping Global Data. The R Journal, 3, 35-43.

**Table 1 Summary statistics and multi-level correlations for person-level measures.**

| | | Variance | | | Correlations | | | | |
|---|---|---|---|---|---|---|---|---|---|
| | Mean | Between | Within | Alpha | 2 | 3 | 4 | 5 | 6 |
| 1. Spatial distancing | 8.60 | 0.21 | 2.17 | 0.74 | 0.43 | 0.44 | 0.02 | 0.15 | −0.02 |
| 2. Physical hygiene | 8.21 | 0.46 | 2.32 | 0.72 | | 0.38 | 0.12 | 0.17 | 0.02 |
| 3. Policy support | 8.29 | 0.94 | 3.45 | 0.81 | | | 0.06 | 0.13 | −0.03 |
| 4. National narcissism | 5.37 | 2.10 | 4.94 | 0.82 | | | | 0.38 | 0.26 |
| 5. National identification | 8.02 | 0.80 | 3.99 | 0.71 | | | | | 0.18 |
| 6. Political ideology | 4.98 | 0.37 | 5.05 | NA | | | | | |

The mean score for each scale is presented along with the variance explained within and between participants and the scale reliability (alpha). There is no alpha for ideology since it is a one-item measure. Higher scores reflect greater support for each measure (and stronger right-wing political beliefs in the case of ideology).

Participants generally reported following the guidelines for contact and hygiene and they supported policies that were intended to reduce the impact of COVID-19 (i.e., means for all three measures were above 8, on scales ranging from 0 to 10; see Table 1). The public health measures were correlated with one another (estimated correlations > 0.38). Consistent with prior work, NI was positively correlated with both NN ($r = 0.38$) and right-wing PI ($r = 0.18$).

We examined relationships between our three measures of socio-political beliefs and COVID preventative behaviours and support of public health policies with a series of multi-level regressions. In these analyses, preventative behaviours and policy support were outcomes, and the three measures of social-political beliefs were modelled simultaneously as predictors. This meant that the relationship between an outcome and each predictor statistically adjusted for relationships between that outcome and the other predictors. The results of these analyses are summarized in Table 2.

NI was significantly and positively related to all public health measures. Individuals with stronger NI (relative to other people within their own nation) reported stronger support for increasing spatial distance and improving physical hygiene and endorsed COVID-19 public health policies more strongly than individuals with weaker identification.

**Table 2 Relationships between outcomes and predictors (including the slope and t-ratio of each relationship). National identification was the strongest predictor of all three COVID-19 public health support measures.**

| Outcome | Predictor | Slope | t-ratio |
|---|---|---|---|
| Spatial distancing | National narcissism | −0.007 | <1 |
| | National identification | 0.129* | 8.63 |
| | Political ideology | −0.028* | 4.44 |
| Physical hygiene | National narcissism | 0.060* | 6.45 |
| | National identification | 0.126* | 11.20 |
| | Political ideology | −0.016 | 2.05 |
| Policy support | National narcissism | 0.029* | 2.89 |
| | National identification | 0.129* | 10.36 |
| | Political ideology | −0.050* | 4.79 |

*$p < 0.001$.

We conducted chi-squares tests comparing the size of these coefficients and found that for all three public health measures, the coefficients for NI were stronger than the coefficients for NN and PI ($ps < 0.001$). Taken together, the three predictors accounted for 8% of the person-level variance of the contact measure, for 8% of the person-level variance of the hygiene

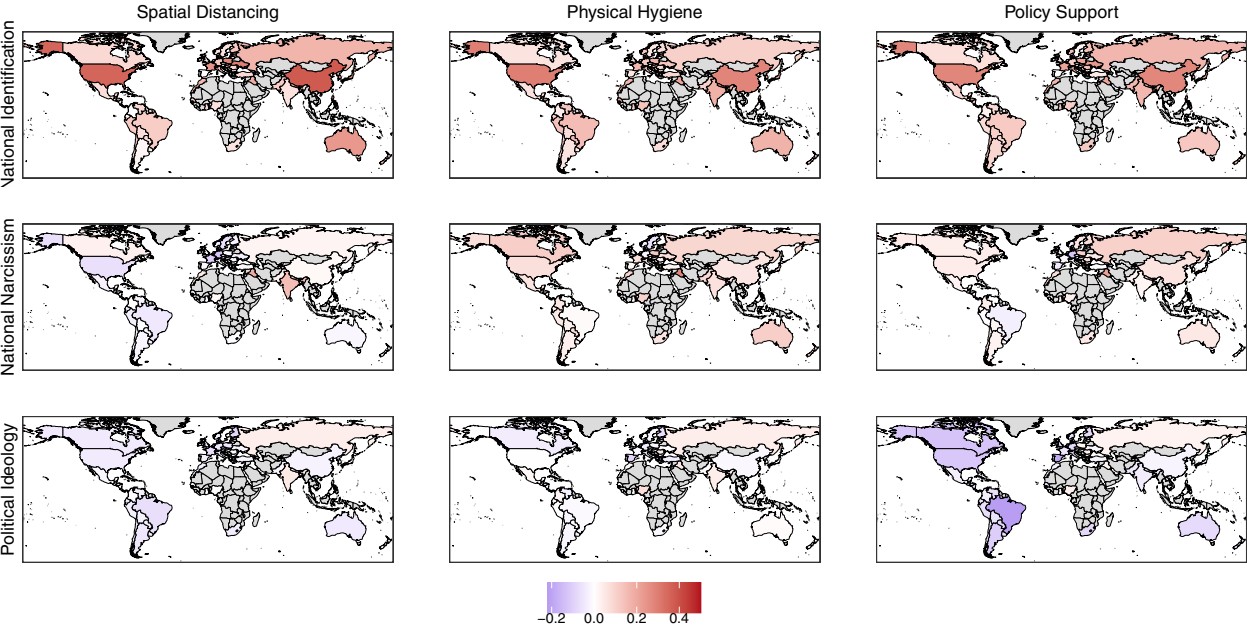

**Fig. 2 Relationships between collective concerns and public health measures in 67 countries and territories.** Heat index depicts the slope coefficients in each country. Blueish colours indicate negative associations between our predictors and our outcomes while reddish colours indicate positive associations (higher scores reflect stronger relationships between national identification, greater national narcissism and greater conservatism, and limiting physical contact, improving hygiene, and supporting public health policies). All the worldmaps were produced using R packages. The map is from the package 'rworldmap' and is licensed-free from South, A. (2011). rworldmap: A New R package for Mapping Global Data. The R Journal, 3, 35-43.

measure, and 5% of the person-level variance of the policy support measure. The coefficients for individual countries are displayed in Figs. 2 and 3. (To see the coefficients and confidences intervals for each variable in each country see Supplementary Figs. 1, 2, and 3).

Study 1 relied on self-report measures. To test the robustness of our predictions, we sought to conceptually replicate our findings using publicly available indices of national identity as well as actual behaviour change during the pandemic in Study 2. To this end, we relied on two publicly available datasets: the World Values Survey[52] and the COVID-19 Google Community Mobility Reports[53] which indicate how people's physical movement has changed in response to COVID-19 policies (available at www.google.com/covid19/mobility/). We created an index of NI using the two relevant items from the World Value Survey (i.e., national pride and closeness to their nation) and an index of physical mobility by averaging community movement across all available places (i.e., retail and recreation, groceries and pharmacies, parks, transit stations, workplaces, and residential). See "Methods" for details about the sample and measures.

We examined whether countries with higher average NI prior to the pandemic predicted a stronger *change* in mobility in response to COVID-19 restrictions during April and May 2020 (This period mirrored when we collected most of the samples in Study 1). We conducted our analysis for the full sample of 42 countries in which aggregate data which was publicly available for both for the NI and the mobility scores.

Replicating the pattern of results from Study 1, NI was associated with reduced spatial mobility, $r = -0.40$, $p = 0.008$ (see Fig. 4; see Supplementary Information for separate correlations for each of the places and the two indices of NIs). The observed association at the aggregate level was moderate to strong. Thus, we found evidence both at the person-level and country-level establishing a link between NI and support for and engagement with public health behaviours.

**Discussion**

Our research suggests that national identities might play an important role in the fight against a global pandemic. Following World War II, early work in social psychology had a tendency to focus on the negative side of nationalism and leadership persuasion, such as destructive obedience to authority[54] and group conformity to incorrect beliefs held by others[55]. In the decades since then, research on social identity[10] and a "social cure" approach to mental health[56] has revealed that there is also a prosocial side to group identity. Based on this latter perspective we predicted, and found, that NI was *positively* associated with support for and engagement with public health behaviours around the globe.

In two global studies combining person-level and country-level analyses, the strength of national identity robustly predicted public health support, operationalized as behavioural health intentions (i.e., physical distance and physical hygiene), support for COVID-19 policy interventions, and reduced physical movement patterns during the pandemic. We found this pattern with self-report measures at the person-level and using measures of actual mobility at the country level. The fact that national identity is associated with large-scale behaviour in real life provides ecologically valid evidence for our main hypothesis. Taken together, these results are consistent with our hypothesis that NI is related to greater behaviour change in compliance with public health policies. We note that the results showing a decline in mobility should be treated with caution, as in the mobility report location accuracy and the categorization of places can vary between countries. In short, people who identified more strongly with their nation reported greater engagement with critical public health measures around the globe.

These results are consistent with the social psychological literature on the benefits of identifying with one's social groups. They also underscore a potential benefit of NI, which might be salient during a national or global health crisis[23]. Our research provides evidence that this form of identification might help to

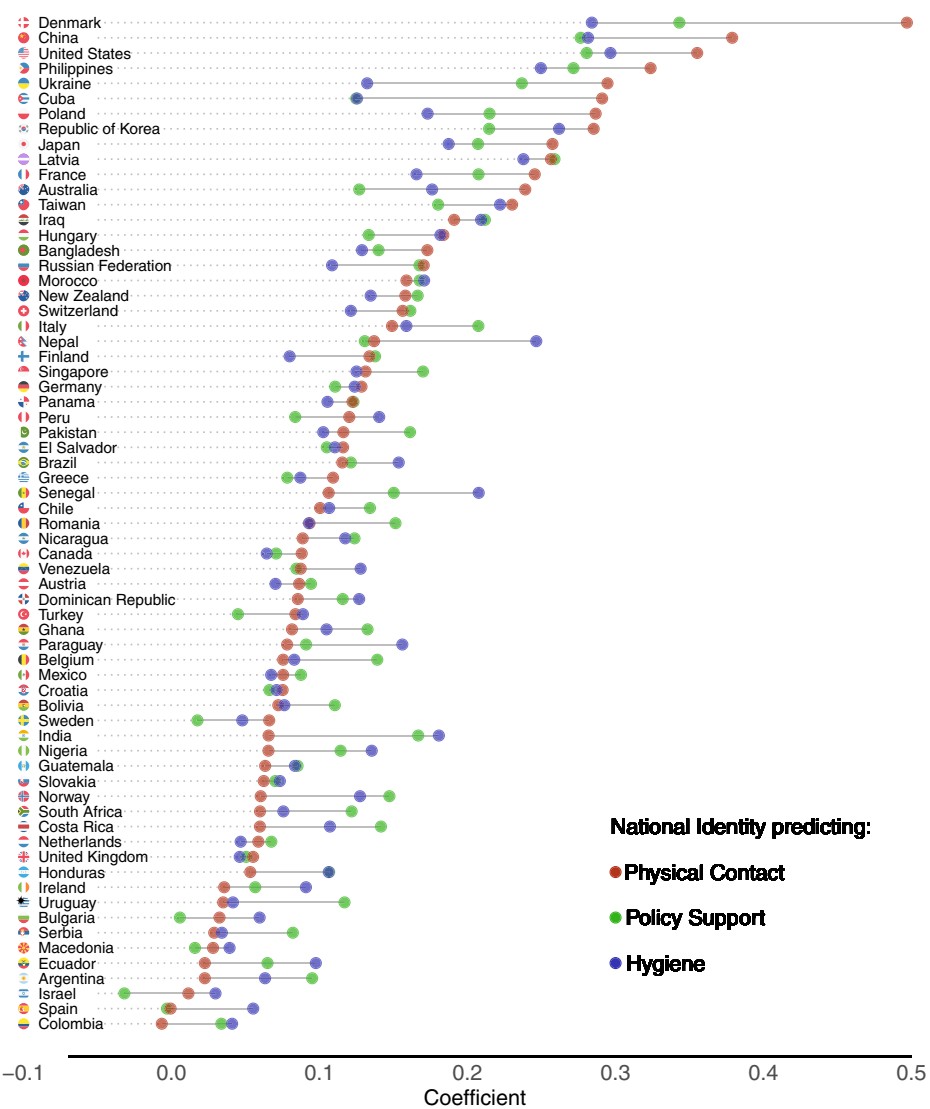

**Fig. 3 Relation between collective concerns and public health measures in 67 countries and territories.** The coefficients reflecting the relation between national identity and each of the health measures are presented for each country from strongest (top) to weakest (bottom). The relation with physical contact (red), policy support (green), and hygiene (blue) are colour coded.

understand public health behaviours. However, work in the United States has found that *threats* to national identity can lead to less support for public health initiatives[57]. As such, mobilizing people around a shared national identity might require considerable nuance. Future work should examine the impact of different types of identity appeals during a pandemic and isolate the causal influence of national identity on real behaviour.

There is reason to believe that other forms of group identification can undercut public health. For instance, partisanship within countries (i.e., when people strongly identify with a specific political party) is associated with risky behaviour[25,26,58]. For example, one study that used geo-tracking data from 15 million smartphones in the US found that counties that voted for a Republican (Donald Trump) over a Democrat (Hillary Clinton) exhibited 14% less spatial distancing during the early stages of the pandemic[26]. These partisan gaps in distancing predicted subsequent increases in infections and mortality in counties that voted for Donald Trump. Moreover, partisanship was a stronger predictor of distancing than many other economic or social factors (e.g., county-level income, population density, religion, age, and state policy). This may be due to leadership, social norms,

and media consumed by people from different identity groups. As such, stronger group identification is not always associated with engagement in public-health behaviour.

It is tempting to conclude that PI might account for these relationships. However, we found that right-wing PI had a positive, moderate correlation with both NI and NN, but very weak correlations with support with public health measures in our multi-country sample. Specifically, right-wing political beliefs were associated with less support for COVID-19 public health policies, compared to left-wing political beliefs. This relationship between political beliefs and compliance has been observed in several countries (e.g., refs. [48,49,59]). Similarly, while NI and NN were associated positively with support for public health measures, right-wing PI was negatively associated with these outcomes. This suggests that a collective identity might be associated with valuing the protection of the entire group during a pandemic, even after adjusting for their ideological differences.

It is also important to note that the relationship between national identity and public health support was distinct from NN. In past research, NN has predominantly been linked to problematic attitudes towards both out-group and in-group

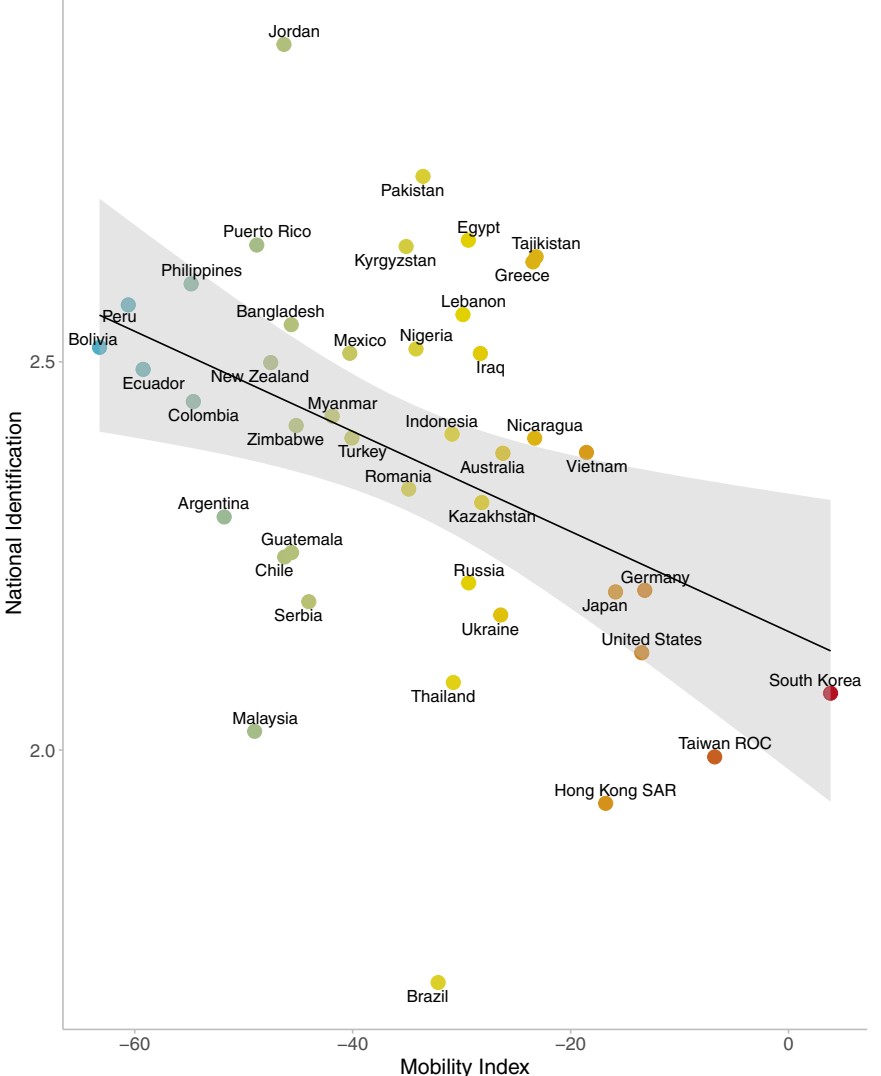

**Fig. 4 Relation between national identification (*y*-axis) and community mobility (*x*-axis) in 42 countries and territories.** Google mobility is depicted as a mean change in mobility during April and May 2020 (i.e., blueish colours indicate a greater reduction of mobility during this period while reddish colours indicate a smaller reduction of mobility). Grey shading is the 95% confidence interval.

members[38,40,60]. However, we found that NN was positively associated with self-reported physical hygiene and support for COVID-19 preventative policies (cf. ref. [42]). Still, these effects were much smaller than those for national identity and depended on the context. Future work should thus carefully consider cross-national differences in human development as well as social norms associated with national identity.

Our evidence suggests that national identity may have modest predictive value for people's endorsement of and adherence to public health measures in the context of a pandemic. This information may be leveraged to create a sense of inclusive nation-based in-groups, potentially increasing engagement with recommended policies. Political and public health leaders might develop effective communication strategies to appeal to a sense of NI. Indeed, this might be particularly helpful in highly polarized countries where adherence to public health recommendations has become a partisan issue (see ref. [26]). For instance, Canadian leaders across the political spectrum adopted similar messaging about the serious risks of the current pandemic which resulted in a rare moment of cross-partisan consensus among the public[61]. Such recategorizations to overarching inclusive national groups (e.g., ref. [62]) may be effective for preventing unhealthy

behaviours. As such, leaders who wish to inspire public health behaviour might benefit from connecting the issue to feelings of national identity. Framing these messages at the level of the nation rather than, for instance, a partisan group, region, or municipality also makes sense when the response requires national coordination[22,63].

However, the effective application of these appeals requires future research as national identity is also implicated in inter-group conflict. This is more likely in the case of NN[36,60], which tends to be associated with lower solidarity with other groups in crisis (e.g., ref. [64]). In the absence of collective narcissism, national identity could reflect not only concerns about protecting one's own country, but also into concern for other nations. Indeed, prior research has found that NI is associated with more positive attitudes towards other nations—especially when adjusting for NN[37,45]. Thus, the nature of national identity might be an important determinant of the effectiveness of identity and the potential for international cooperation. In addition, it could turn out that a commitment to cosmopolitanism or other supranational identities and ideologies may play a role that bolsters what we have seen in the case of national identity[65].

One major strength of our paper is the scope of nations we included in our samples. The first study included data from 67 nations and the second study included data from 42 countries. The vast majority of published research in psychology and social sciences has been conducted in so-called WEIRD cultures[66], typically restricted to the narrow **w**estern and **e**ducational setting of American or European university students, and non-representative participants from **i**ndustrialized, **r**ich and **d**emocratic countries. The COVID-19 pandemic, however, is a truly *global* issue underscoring the importance of gathering samples outside these WEIRD cultures. Moreover, it was striking to see that the same person-level association between NI and our public health measures was in the same direction in almost every country we studied. Although we managed to collect data from a wide variety of countries and territories, we were unable to obtain samples from every nation (especially in Africa and the middle east). As such, we encourage future research in these countries to see if the same dynamics are at play.

Another element of our paper was an attempt to collect representative or stratified samples in Study 1. While most studies in psychology focus on convenience samples (e.g., undergraduate or MTurk participants), it is important to gather samples that are more diverse with regards to gender, age, and other key risk factors during a pandemic. Collecting representative samples affords the opportunity to help make better generalizations to the wider population within each country as well as the broader sample of countries around the globe. Due to funding constraints, we were not able to obtain representative samples from most nations. As such, we are unable to make strong generalizations about the populations in those countries. But note that we did directly compare the findings in more vs. less representative samples and found no significant difference in the overall relationship between NI and all three public health measures (see Supplementary Information for details).

This research was correlational and conducted during the early phase of the pandemic. Although a causal relation between NI and public health behaviour makes sense from a theoretical perspective, we cannot rule out the possibility that public health behaviour causes NI, or that both are caused by a third variable (e.g., ref. [23]). Moreover, we have no evidence whether this pattern would apply during later stages of the current or future pandemics. Indeed, national identity may increase during times of crisis as people recognize their duty as citizens to help respond to this issue. We encourage future work to experimentally manipulate the salience of NI or frame health messages in a way that highlights the link between identification and the public health measures. Another limitation is the exclusive focus on national groups rather than, for instance, identification with a city, region, religion, or ethnic group—or, for that matter, all of humanity. Some research suggests that local leaders may be ineffective if their advice contradicts a national leader (see ref. [26]). In the current pandemic, nations have been among the most important actors for implementing policy or promoting national health guidelines, but sub-national units and international organizations such as the World Health Organization also play an important role.

The COVID-19 pandemic spreading across the world is one of the most devastating global health crises of the past century. Until a verifiably safe and effective vaccine or therapeutic treatment is universally administered, efforts to inspire collective action for greater compliance with public health measures remain a central challenge when mitigating the transmission of the SARS-CoV-2 virus (e.g., spatial distancing, physical hygiene, and support for health policies). Moreover, understanding social identity and collective behaviour likely plays a key role in vaccination efforts[67]. Our large-scale studies suggest that identification with one's nation is positively associated with support for and engagement in critical behavioural public health measures. Understanding the role of social identity appears to be an important issue when addressing public health crises.

## Methods

In Study 1, we launched a call using social media to collect data all over the world on psychological factors that might be related to COVID-19 pandemic response, with public health support as the primary outcome in April 2020. Each team was asked to collect data from at least 500 participants, representative with respect to gender and age, in their own country or territory. We created a survey in English (see below) that we sent to each team. The survey was approved by the ethics board at the University of Kent (each research team was allowed to include additional items after the main survey under their own institutional protocol). We have complied with all relevant ethical regulations and all participants were asked to give informed consent. Where necessary, each team translated the survey into the local language, using the standard forward-backward translation method, and then collected the data. The datasets were then collated and analyzed using multi-level models. We report how we determined our sample size, all data exclusions (if any), all manipulations, and all measures in the study (see Supplementary Information). All materials and data are available at: https://osf.io/y7ckt/.

Raw data we obtained from all collaborators were cleaned to exclude any duplicate answers as well as those younger than 18 years or older than 100 years. We then excluded data from two participants from Puerto Rico and 313 participants recruited from the UEA where it was difficult to establish participant nationality. This resulted in a sample of 51,089 participants. For the current analysis, we also excluded participants who had missing data on all six key variables of interest. We were left with a sample of 49,968 for analyses (Mean age = 43; Gender = 52% females). Figure 1 shows the geographical distribution of countries included in the project (For a full list and sample characteristics from each country, please see Supplementary Information). The sample includes countries from all continents except for Antarctica. Due to our open call for collaborators, some continents are overrepresented (e.g., Europe, Americas) while others are underrepresented (e.g., Africa, Middle East).

We encouraged teams to collect nationally representative samples. Of the 67 countries in which data were collected, representative samples were collected in 28, convenience samples were collected in 36, and both types of sampling were used in three countries. To determine if the relationships that were the focus of our paper varied as a function of the type of sample, we conducted analyses that compared coefficients for countries that had the three types of samples. These analyses found only one difference as a function of type of sample. Type of sample moderated the slope between spatial distancing and national identity. The overall mean slope was 0.12, and the estimated slope for countries that collected representative samples was 0.16, whereas it was 0.08 for countries that collected convenience samples. Importantly, both were statistically significant from 0 ($p < 0.001$).

Questionnaires were administered online. Each participant completed a series of psychological measures and self-reported public health behaviours (see complete survey with all items in Supplementary Information). Participants completed the scales in random order.

For the current paper, we focused on three potential predictors of public health support. Our primary predictor was a two-item NI measure (which included one item from ref. [9] and an additional item measuring identity centrality from ref. [8]): "I identify as (nationality)" and "Being a (nationality) is an important reflection of who I am". Our secondary predictor was a three-item *NN* scale[36], which included the following sample item: "My (national group) deserves special treatment." The nationalities were provided by the survey researchers. These measures used an 11-point slider scale with three labels items: 0 = *"strongly disagree"*, 5 = *"neither agree nor disagree"*, 10 = *"strongly agree"*.

As a third predictor, we included a one-item measure of *PI*: "Overall, how would you best describe yourself in terms of PI?". This measure used a scale from 0 = *extremely liberal/left-leaning* to 10 = *extremely conservative/right-leaning*. This single-item measure of ideology has been found to account for a significant proportion of the variance in presidential voting intentions in American National Election studies between 1972 and 2004[68]. We included the terms left-leaning and right-leaning to make the item generalizable to numerous political systems.

As the primary outcome variable, we included three measures of public health support. A *spatial distancing* scale, consisting of five items, as, for example, "During the days of the coronavirus (COVID-19) pandemic, I have been staying at home as much as practically possible". (Prior to conducting our analyses, we learned that the five-item scale had low reliability ($\alpha = 0.002$). However, after dropping one bad item the scale had acceptable reliability ($\alpha = 0.72$). As such, all analyses reported in the paper use this four-item version of the scale.) A *physical hygiene* scale, consisting of five items, as, for example, "During the days of the coronavirus (COVID-19) pandemic, I have been washing my hands longer than usual". A *policy support* scale, consisting of five items, as, for example, "During the days of the coronavirus (COVID-19) pandemic, I have been in favour of closing all schools and universities". We used an 11-point "slider scale with three labels: 0 = *"strongly disagree"*, 50 = *"neither agree nor disagree"*, 100 = *"strongly agree"*, which was re-coded to a scale from 0 to 10.

To see if these relationships varied as a function of socio-economic factors and the state of the pandemic in each country, we examined several country-level factors. Specifically, we included the 2018 (most recent available) HDI (ranging from 0 to 1), which represents a combined index of life expectancy at birth, level of education (mean years of schooling for adults over 25 and expected years of schooling for children), and national wealth (gross national income per capita[69]).

To ensure our results were not confounded with the pandemic rate across countries, we measured the total COVID-19 infection and mortality cases (as well as the infection and mortality rate per capita) in each country at the start of data collection for this project. Our main findings did not vary as a function of total infections and deaths as well as infections and deaths per capita at the start of data collection for this project[70] (April 17, 2020). These variables had very little impact on the results and are not discussed further. All measures will be made publicly available upon publication at the *Open Science Framework* website.

We conceptualized the data as a multi-level data structure in which persons were nested within countries, and we analyzed the data with a series of multi-level models (MLM) using the programme HLM[71] (see ref. [51] for a description of using MLM to analyze data from multinational studies). The analyses examined within-country (person-level) relationships between behavioural health-protective responses to COVID-19 (i.e., spatial distancing, physical hygiene, and policy support) and individual differences in collective concerns (i.e., NI, NN, and PI). We also examined the moderating effects of country-level differences on these person-level relationships. For instance, we examined if these person-level relationships between collective concerns and health-protective measures varied as a function of between-country differences in overall human development as measured by the HDI or national rates of COVID-19 infections and mortality.

Before examining relations between COVID-19 protection and socio-political attitudes, we examined the reliability of our measures (with the exception of PI, which was measured with only one item). These analyses consisted of models in which the i items in a scale were nested within j persons, which were nested within k countries. Such analyses provide the multi-level equivalent of a Cronbach's alpha[72,73]. The model is below.

Level 1 (item level): $y_{ijk} = \pi_{0jk} + e_{ijk}$
Level 2 (person-level): $\pi_{0jk} = b_{00k} + r_{0jk}$
Level 3 (country-level): $b_{00k} = g_{000} + u_{00k}$

In the level 1 model, $y_{ijk}$ is response $i$, for person $j$, in country $k$, $\pi_{0jk}$ is a random coefficient representing the mean response for person $j$ in country $k$, $b_{0j}$ is a random coefficient representing the mean of $y$ for country $k$ (across the j persons in each country), $e_{ijk}$ represents the error associated with each measure, and the variance of $r_{ijk}$ constitutes the within-country variance. In multi-level modelling, the coefficients from one level of analysis are passed up to the next. In the level 3 model, $g_{000}$ represents the grand mean of the country-level means ($b_{00k}s$) from the person-level model, $u_{00k}$ represents the error of $b_{00k}$, and the variance of $u_{00k}$ constitutes the level 3, country-level variance.

These analyses suggested that, with the exception of spatial distancing, our scales were at least "moderately" reliable[74] ($\alpha > 0.60$). The reliability estimates and descriptive statistics are presented in Table 1. For spatial distancing, follow-up analyses indicated that a reliable scale could be created from items 1, 3, 4, and 5. Item 2 asking about visiting friends, family or colleagues was therefore dropped from the final analyses.

The estimated means suggest that people generally reported following the guidelines for contact and hygiene and they supported policies that were intended to reduce the impact of COVID-19 (i.e., means for all three measures were above 8, on scales ranging from 0 to 10). Moreover, although the majority of variance in NI, NN, and PI was within-country, there was also notable between-country variance. This justified further analyses of relations between country-level means of these measures and HDI. We calculated scale means and used Mplus[75] to estimate multi-level correlations for person-level measures, controlling for the nested structure of the data (see Table 1).

The next set of analyses examined relations between scores on the HDI and the means of the person-level measures. This model was a variant of the unconditional model. HDI scores were entered as a predictor in the country-level model presented above (level 3). MLM analyses do not estimate standardized coefficients, and to simplify the interpretation of the results, HDI scores were standardized prior to analysis (and, therefore, were entered uncentered). Note that these analyses account for the reliability of scales. By nesting items within persons, we estimated a latent mean for each construct.

The results of these analyses are presented in Table 2. For all measures, except PI, there were negative relationships between HDI scores and country-level means. Note that the coefficients in the table represent the change in a country-level mean associated with a 1SD increase in HDI scores. In other words, citizens in countries with higher scores on the global HDI also reported less support for COVID-19 public health measures. Effect sizes are defined as the percent reduction in the country-level variance of a null model (Table 2) associated with the inclusion of HDI scores at the country level. Because PI was measured with only one item, the variance estimates and effect size for PI are from a two-level model (persons nested within countries). Estimating effect sizes for multi-level analyses such as those used in the present study are discussed in Nezlek[51].

Next, we examined person-level relationships between the three COVID-19 protection measures (modelled as outcomes) and NI, NN, and PI (modelled as predictors). Predictors were defined as the mean scores for each scale. To account

for relationships among the predictors, all predictors were entered at the person level of the model. Predictors were entered group-mean centred and were modelled as randomly varying. Again, because this was done using a three-level model in which the first level was a measurement level, outcomes were modelled as latent means.

Entering predictors group-mean centred meant that estimates of coefficients controlled for country-level differences in means[51]. Entering predictors as randomly varying meant that the model account for the possibility that slopes varied between countries. In essence, a regression equation, consisting of an intercept and a set of slopes, was estimated for each country, and these estimates were "passed up" to the country level where they were tested for significance. The model is below (item level is not shown).

Level 2 (person-level): $\pi_{0jk} = b_{00k} + b_{01k}*(NN) + b_{02k}*(NI) + b_{03k}*(PI) + r_{0jk}$
Level 3 (intercept): $b_{00k} = g_{000} + u_{00k}$
Level 3 (NN slope): $b_{00k} = g_{010} + u_{01k}$
Level 3 (NI slope): $b_{00k} = g_{020} + u_{02k}$
Level 3 (PI slope): $b_{00k} = g_{030} + u_{03k}$

The hypothesis of interest was tested by assessing the significance of the $g_{010}$, $g_{020}$, and $g_{030}$ coefficients in this model. Was the mean slope between an outcome and a predictor significantly different from 0? These unstandardized coefficients represent the expected change in an outcome for a one-unit increase in a predictor, i.e., an increase of one on a scale (out of 11). Also, the random error terms for all predictors were significant at $p < 0.001$.

According to these analyses, NI was the most reliable and strongest predictor of our COVID-19 public health support measures (see Fig. 2 for the coefficients in each country as well as Supplementary Figures 1, 2, and 3 for the coefficients with 95% confidence intervals). It was significantly and positively related to all three measures (even after adjusting for NN and PI). Individuals with stronger NI (relative to other people within their own nation) reported stronger support for limiting physical distance and improving physical hygiene than individuals with weaker identification, and they also endorsed COVID-19 public health policies to a greater extent.

NN was significantly positively related to two of the three protective measures (albeit weakly). Individuals scoring higher in NN supported recommendations for physical hygiene and endorsed COVID-19 related policies more strongly compared to individuals with lower levels of NN.

The relationships between PI and public health support were negative (albeit weakly) for all three measures, indicating that individuals with more left-leaning or liberal political orientation tended to endorse recommendations for contact and hygiene and supported COVID-19-related policies more strongly than those with more right-leaning or conservative political orientation.

Effect sizes were estimated using a similar procedure to that used for estimating effect sizes at the country-level. Effect sizes were defined as the percent reduction in the person-level variance of a null model (Table 2) associated with the inclusion of the three predictors (collective narcissism, NI, and PI) at the person level. The three predictors accounted for 8% of the person-level variance of the contact measure, for 7% of the person-level variance of the hygiene measure, and 5% of the person-level variance of the policy support measure.

Next, we modelled country-level factors, such as the HDI to examine whether the relations between person-level factors, like NI, and public health support would remain after adjusting for the general health and standard of living in a country. The HDI is a measure of achievement in key dimensions of human development: a long and healthy life, being knowledgeable, and having a decent standard of living. The HDI is the mean of normalized indices for each of the three dimensions (see ref. [76]). Specifically, we examined if person-level relations (slopes) between collective concerns and COVID-19 public health support varied across countries as a function of HDI by adding HDI scores to the level 3 model that examined slopes. The relationships between NI and each of the three public health measures were not heavily impacted or moderated by HDI. Indeed, we observed only two modest moderating effects.

We found that HDI moderated the relationships between NN and spatial distancing ($g_{011} = -0.03$, $t = 2.93$, $p < 0.01$). The relationship between NN and spatial distancing was negative in countries that had higher HDI scores (the estimated slope for a country +1 SD on the HDI was 0.037) but positive in countries that had lower HDI scores (the estimated slope for a country −1 SD on the HDI was 0.027). We also found that HDI moderated the relationship between PI and hygiene ($g_{031} = -0.016$, $t = 2.16$ $p = 0.034$). The overall negative relationship between right-wing PI and hygiene was stronger in countries that had higher HDI scores (the estimated slope for a country +1 SD on the HDI was −0.031) than in countries that had lower HDI scores (the estimated slope for a country −1 SD on the HDI was 0.002, functionally 0). We note that these effects were not statistically significant at the $p < 0.001$ threshold we used for Study 1 so we recommend interpreting them with caution.

In Study 2, we accessed data from two publicly available datasets: the World Values Survey[52] and the COVID-19 Google Community Mobility Reports[53] which indicate how people's physical movement has changed in response to COVID-19 policies (available at www.google.com/covid19/mobility/). We examined whether countries with higher average NI would also show stronger *change* in mobility in response to COVID-19 restrictions during April and May 2020. We created an index of NI using the two relevant items from the World Value Survey (i.e., national pride and closeness to their nation) and an index of physical mobility by

averaging community movement across all available places (i.e., retail and recreation, groceries and pharmacies, parks, transit stations, workplaces, and residential). We analyzed all 42 countries in which aggregate data was publicly available for both for NI and mobility scores. The study was approved by the ethics board at the University of Kent. All materials and data are available at: https://osf.io/y7ckt/.

NI was computed based on indices from the first release of data from Wave 7 of the World Value Survey. The surveys were conducted between early 2017 to mid-2020. All countries employed random probability representative samples of the adult population (We computed country averages using default weights applied in the World Values Survey dataset. However, our results are very similar whether are not these weights are applied). Our analysis focused on two indices. First, we used the national pride question: "How proud are you to be [country's nationality]? 1 = Very proud, 2 = Quite proud, 3 = Not very proud, 4 = Not at all proud, and 5 = I am not [country's nationality]. (In some countries, this source item actually refers to pride of "being a citizen [of the country]." A response choice was available for respondents who were not citizens of the country where they were interviewed in Wave 7 of the World Value Survey. While some countries differ in terms of their ethnic or civic-based notions of citizenship, we used NI to denote overall identification with a specific national polity.) We excluded the latter category and re-coded the remaining responses on a scale from 0 = Not at all proud to 3 = Very proud.

The second item captured closeness to one's country: "People have different views about themselves and how they relate to the world. Using this card, would you tell me how close do you feel to [country]?" 1 = Very close, 2 = Close, 3 = Not very close, 4 = Not close at all. We re-coded the responses on a scale from 0 = Not close at all proud to 3 = Very close. (Note that participants can refuse to respond or indicate "I don't know" to both items. These responses were coded as missing.) The two items were positively correlated at country-level ($r = 0.31$, $p = 0.049$), so we averaged them to create a composite index of NI ($M = 2.38$, $SD = 0.24$).

Community mobility was computed based on Google Community Mobility Reports, which indicate how people's aggregate physical movement has changed over time. The reports show movement trends over time across different categories of places: retail and recreation, groceries and pharmacies, parks, transit stations, workplaces, and residential. Percentage change for each day is computed relative to a baseline, which is a median value, for the corresponding day of the week, during the 5-week period from Jan 3 to Feb 6, 2020. To create our overall index of reductions in community mobility, we computed average indices for each of the places over April and May 2020 (to roughly match the time frame of Study 1). We then created a composite index of mobility by averaging mobility across all places, with residential mobility reverse-coded ($\alpha = 0.91$, $M = -34.87$, $SD = 15.15$). This translates to a 35% reduction in movement from the start of the calendar year to the spring in these 42 nations.

**Reporting summary**. Further information on research design is available in the Nature Research Reporting Summary linked to this article.

## Data availability

The data generated during and/or analyzed during the current study are available on the Open Science Framework repository, https://osf.io/y7ckt/. The publicly available datasets that support the results of this study, The World Values Survey and the COVID-19 Google Community Mobility Reports, are available from https://www.worldvaluessurvey.org/wvs.jsp, and www.google.com/covid19/mobility/, respectively.

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

## Acknowledgements

The authors wish to thank Katie Eilish Brown for constructive comments throughout the editorial process. We also acknowledge the following funding sources: John Templeton Foundation (JTF) - 61378 [Van Bavel] Narodowe Centrum Nauki (National Science Centre) - 2018/29/B/HS6/02826 [Cislak] RCUK│Medical Research Council (MRC) - MR/P014097/1 [Lockwood] Economic Social Research Council Impact Acceleration Award, University of Oxford [Lockwood] Gouvernement du Canada │ Social Sciences and Humanities Research Council of Canada (Conseil de recherches en sciences humaines du Canada) - 130760 [Choma] Gouvernement du Canada │ Social Sciences and Humanities Research Council of Canada (Conseil de recherches en sciences humaines du Canada) - SSHRC-506547 [Cunningham] Agentúra na Podporu Výskumu a Vývoja (Slovak Research and Development Agency) - APVV-17-0596 [Findor] Narodowe Centrum Nauki (National Science Centre) - 2015/19/B/HS6/01253 [Jasko] Academy of Finland (Suomen Akatemia) [Laakasuo] Austrian Science Fund (Fonds zur Förderung der Wissenschaftlichen Forschung) - I3381 [Lamm] Universität Wien (University of Vienna) [Lamm] Ministry of Science and Technology, Taiwan (Ministry of Science and Technology of Taiwan) [Lin] Aarhus Universitets Forskningsfond (Aarhus University Research Foundation) - AUFF-E-201 9-9-4 [Mitkidis] Vetenskapsrådet (Swedish Research Council) - 2018-00877 [Olsson] Aarhus Universitets Forskningsfond (Aarhus University Research Foundation) - 28207 [Otterbring] Carlsbergfondet (Carlsberg Foundation) - CF20-0044 [Petersen] Ministarstvo Prosvete, Nauke i Tehnološkog Razvoja (Ministry of Education, Science and Technological Development of the Republic of Serbia) - 47010 [Todosijević] NOMIS Stiftung (NOMIS Foundation) [Tsakiris] Ministry of Science and Technology, Taiwan (Ministry of Science and Technology of Taiwan) [Tung] National Natural Science Foundation of China (National Science Foundation of China) - 71972065 [Zhang] National Natural Science Foundation of China (National Science Foundation of China) - 71602163 [Zhang] RCUK│Biotechnology and Biological Sciences Research Council (BBSRC) - BB/R010668/1 [Apps] Agence Nationale de la Recherche (French National Research Agency) - ANR-17-EURE-0010 [Conway] Gouvernement du Canada │ Social Sciences and Humanities Research Council of Canada (Conseil de recherches en sciences humaines du Canada) - SSHRC-506547 [Davis] Deutsche Forschungsgemeinschaft (German Research Foundation) - EXC 2052/1 – 390713894 [Frempong] Gouvernement du Canada │ Natural Sciences and Engineering Research Council of Canada (Conseil de Recherches en Sciences Naturelles et en Génie du Canada) [Fugelsang] Agentúra na Podporu Výskumu a Vývoja (Slovak Research and Development Agency) - APVV-17-0596 [Hruška] Carlsbergfondet (Carlsberg Foundation) - CF20-0044 [Jørgensen] Gouvernement du Canada │ Social Sciences and Humanities Research Council of Canada (Conseil de recherches en sciences humaines du Canada) - SSHRC-506547 [Long] Austrian Science Fund (Fonds zur Förderung der Wissenschaftlichen Forschung) - I3381 [Nitschke] Universität Wien (University of Vienna) [Nitschke] Deutsche Forschungsgemeinschaft (German Research Foundation) - EXC 2052/1 – 390713894 [Stadelmann] Agence Nationale de la Recherche (French National Research Agency) - ANR-10-IDEX-0001-02 PSL [Strickland] Agence Nationale de la Recherche (French National Research Agency) - ANR-17-EURE-0017 [Strickland] Vetenskapsrådet (Swedish Research Council) [Tinghög] HKUST IEMS research grant

project, funded by EY [Tyrala] Vetenskapsrådet (Swedish Research Council) [Västfjäll] Gouvernement du Canada | Social Sciences and Humanities Research Council of Canada (Conseil de recherches en sciences humaines du Canada) - 435-2012-1135 [Wohl] Coordenação de Aperfeiçoamento de Pessoal de Nível Superior (Brazilian Federal Agency for the Support and Evaluation of Graduate Education) - 88887.310255/2018 [Boggio] Coordenação de Aperfeiçoamento de Pessoal de Nível Superior (Brazilian Federal Agency for the Support and Evaluation of Graduate Education) - 1133/2019 [Boggio] Ministry of Science, Technology and Innovation | Conselho Nacional de Desenvolvimento Científico e Tecnológico (National Council for Scientific and Technological Development) - 309905/2019-2 [Boggio] Research Council of Norway through its Centres of Excellence Scheme, FAIR project No 262675 [Sjåstad] Institute of Social Sciences Ivo Pilar [Pavlović] J. William Fulbright Program [Azevedo] Institute for Lifecourse Development, University of Greenwich [Birtel] Institute of Social Sciences Ivo Pilar [Franc] Project Pro.Co.P.E., IMT School (PAI2019) [Bilancini] Italian Ministry of University and Research - PRIN 2017 (20178293XT) [Boncinelli] Princeton Graduate Student Research Funding (Program in Cognitive Science) [Vlasceanu] Corruption Laboratory on Ethics, Accountability, and the Rule of Law (CLEAR), University of Virginia [Yucel] Institute for Lifecourse Development, University of Greenwich [Farmer] Research Council of Norway through its Centres of Excellence Scheme, FAIR project No 262675 [Ay] Charles Koch Foundation, Center for the Science of Moral Understanding [Gray] Australian Research Council (DP180102384) [Levy] JSPS KAKENHI (JP16H03079, JP17H00875, JP18K12015, JP20H04581, and 21H03784) [Yamada] St Andrews and Stirling Graduate Programme Research Funding [Schönegger] Institute of Social Sciences Ivo Pilar [Maglić] São Paulo Research Foundation - FAPESP (2019/27100-1) [Sampaio] Institute of Social Sciences Ivo Pilar [Mikloušić] Seele Neuroscience Social Projects Fund (2020/004) [Monroy-Fonseca] São Paulo Research Foundation - FAPESP (2019/26665-5) [Rego]

## Author contributions

Jay J. Van Bavel launched the project, oversaw the design of the study, data collection, curation and analysis, and wrote the manuscript. Aleksandra Cichocka and Paulo S. Boggio oversaw the design of the study, data collection, curation, and analysis, conducted data analysis, and wrote the manuscript. Aleksandra Cichocka coordinated ethics approval. Valerio Capraro and Hallgeir Sjåstad designed the study, oversaw data collection, and contributed to writing the manuscript. John B. Nezlek, Flavio Azevedo, Tomislav Pavlović, Gabriel G. Rêgo, and Waldir M. Sampaio conducted data analysis. Flavio Azevedo coordinated data curation efforts. Flavio Azevedo, Tomislav Pavlović, Gabriel G. Rêgo, and Waldir M. Sampaio prepared the dataset and its metadata. Flavio Azevedo, Tomislav Pavlović, Gabriel G. Rêgo and Aleksandra Cichocka organised project documentation. Flavio Azevedo designed, populated, and maintained the ICSMP project's website. Bjarki Gronfeldt, Anni Sternisko, Hans H. Tung and Ming-Jen Lin assisted with data preparation, documentation and analysis. Mark Alfano, Michele J. Gelfand, Flavio Azevedo, Michèle D. Birtel, Aleksandra Cislak, Claus Lamm, Patricia L. Lockwood, Robert Malcolm Ross, Biljana Gjoneska, and F. Ceren Ay contributed to various phases of the organisation, including writing the manuscript. Waqas Ejaz, Annalisa Myer, and Valerio Capraro were responsible for creating the authors' list, as well as the contributions statement. Valerio Capraro, Koen Abts, Elena Aguadullina, John Jamir Benzon Aruta, Sahba Nomvula Besharati, Alexander Bor, Becky L. Choma, Charles David Crabtree, William A. Cunningham, Koustav De, Waqas Ejaz, Christian T. Elbaek, Andrej Findor, Daniel Flichtentrei, Renata Franc, Biljana Gjoneska, June Gruber, Estrella Gualda, Yusaku Horiuchi, Toan Luu Duc Huynh, Agustin Ibanez, Mostak Ahamed Imran, Jacob Israelashvili, Katarzyna Jasko, Jaroslaw Kantorowicz, Elena Kantorowicz-Reznichenko, André Krouwel, Michael Laakasuo, Claus Lamm, Caroline Leygue, Ming-Jen Lin, Mohammad Sabbir Mansoor, Antoine Marie, Lewend Mayiwar, Honorata Mazepus, Cillian McHugh, John Paul Minda, Panagiotis Mitkidis, Andreas Olsson, Tobias Otterbring, Dominic J. Packer, Anat Perry, Michael Bang Petersen, Arathy Puthillam, Julián C. Riaño-Moreno, Tobias Rothmund, Hernando Santamaría-García, Petra C. Schmid, Drozdstoy Stoyanov, Shruti Tewari, Bojan Todosijević, Manos Tsakiris, Hans H. Tung, Radu G. Umbreș, Edmunds Vanags, Madalina Vlasceanu, Andrew Vonasch, Meltem Yucel, Yucheng Zhang acted as national team leaders and were responsible for data collection in their own country. Mohcine Abad, Eli Adler, Narin Akrawi, Hamza Alaoui Mdarhri, Hanane Amara, David M. Amodio, Benedict G. Antazo, Matthew Apps, F. Ceren Ay, Mouhamadou Hady Ba, Sergio Barbosa, Brock Bastian, Anton Berg, Maria P. Bernal-Zárate, Michael Bernstein, Michał Białek, Ennio Bilancini, Natalia Bogatyreva, Leonardo Boncinelli, Jonathan E. Booth, Sylvie Borau, Ondrej Buchel, Chrissie F. Carvalho, Tatiana Celadin, Chiara Cerami, Hom Nath Chalise, Xiaojun Cheng, Luca Cian, Kate Cockcroft, Jane Conway, Mateo Andres Córdoba-Delgado, Chiara Crespi, Marie Crouzevialle, Jo Cutler, Marzena Cypryańska, Justyna Dabrowska, Michael A. Daniels, Victoria H. Davis, Pamala N Dayley, Sylvain Delouvee, Ognjan Denkovski, Guillaume Dezecache, Nathan A. Dhaliwal, Alelie B. Diato, Roberto Di Paolo, Marianna Drosinou, Uwe Dulleck, Jānis Ekmanis, Arhan S. Ertan, Tom W Etienne, Hapsa Hossain Farhana, Fahima Farkhari,

Harry Farmer, Ali Fenwick, Kristijan Fidanovski, Terry Flew, Shona Fraser, Raymond Boadi Frempong, Jonathan A. Fugelsang, Jessica Gale, E. Begoña Garcia-Navarro, Prasad Garladinne, Oussama Ghajjou, Theofilos Gkinopoulos, Kurt Gray, Siobhán M. Griffin, Bjarki Gronfeldt, Mert Gümren, Ranju Lama Gurung, Eran Halperin, Elizabeth Harris, Volo Herzon, Matej Hruška, Guanxiong Huang, Matthias F. C. Hudecek, Ozan Isler, Simon Jangard, Frederik Juhl Jørgensen, Frank Kachanoff, John Kahn, Apsara Katuwal Dangol, Oleksandra Keudel, Lina Koppel, Mika Koverola, Emily Kubin, Anton Kunnari, Yordan Kutiyski, Oscar Laguna, Josh Leota, Eva Lermer, Jonathan Levy, Neil Levy, Chunyun Li, Elizabeth U. Long, Chiara Longoni, Marina Maglić, Darragh McCashin, Alexander L Metcalf, Igor Mikloušić, Soulaimane El Mimouni, Asako Miura, Juliana Molina-Paredes, César Monroy-Fonseca, Elena Morales-Marente, David Moreau, Rafał Muda, Annalisa Myer, Kyle Nash, Tarik Nesh-Nash, Jonas P. Nitschke, Matthew S. Nurse, Yohsuke Ohtsubo, Victoria Oldemburgo de Mello, Cathal O'Madagain, Michal Onderco, M. Soledad Palacios-Galvez, Jussi Palomäki, Yafeng Pan, Zsófia Papp, Philip Pärnamets, Mariola Paruzel-Czachura, Zoran Pavlović, César Payán-Gómez, Silva Perander, Michael Mark Pitman, Rajib Prasad, Joanna Pyrkosz-Pacyna, Steve Rathje, Ali Raza, Gabriel G. Rêgo, Kasey Rhee, Claire E. Robertson, Iván Rodríguez-Pascual, Teemu Saikkonen, Octavio Salvador-Ginez, Waldir M. Sampaio, Gaia C. Santi, Natalia Santiago-Tovar, David Savage, Philipp Schönegger, David T. Schultner, Enid M. Schutte, Andy Scott, Madhavi Sharma, Pujan Sharma, Ahmed Skali, David Stadelmann, Clara Alexandra Stafford, Dragan Stanojević, Anna Stefaniak, Anni Sternisko, Augustin Stoica, Kristina K. Stoyanova, Brent Strickland, Jukka Sundvall, Jeffrey P. Thomas, Gustav Tinghög, Benno Torgler, Iris J. Traast, Raffaele Tucciarelli, Michael Tyrala, Nick D. Ungson, Mete S. Uysal, Paul A. M. Van Lange, Jan-Willem van Prooijen, Dirk van Rooy, Daniel Västfjäll, Peter Verkoeijen, Joana B. Vieira, Christian von Sikorski, Alexander Cameron Walker, Jennifer Watermeyer, Erik Wetter, Ashley Whillans, Robin Willardt, Michael J. A. Wohl, Adrian Dominik Wójcik, Kaidi Wu, Yuki Yamada, Onurcan Yilmaz, Kumar Yogeeswaran, Carolin-Theresa Ziemer, Rolf A. Zwaan were involved in the translation of the survey in their local language and in data collection. All authors revised and approved the final manuscript.

## Funding

## Competing interests

André Krouwel (ownership and stocks in Kieskompas BV, data collector in this project). No payment was received by the author. No other authors reported a competing interest.

## Additional information

Jay J. Van Bavel[1✉], Aleksandra Cichocka[2], Valerio Capraro[3], Hallgeir Sjåstad[4], John B. Nezlek[5,6], Tomislav Pavlović[7], Mark Alfano[8], Michele J. Gelfand[9], Flavio Azevedo[10✉], Michèle D. Birtel[11], Aleksandra Cislak[5], Patricia L. Lockwood[12,13], Robert Malcolm Ross[14], Koen Abts[15], Elena Agadullina[16], John Jamir Benzon Aruta[17], Sahba Nomvula Besharati[18], Alexander Bor[19], Becky L. Choma[20], Charles David Crabtree[21], William A. Cunningham[22], Koustav De[23], Waqas Ejaz[24], Christian T. Elbaek[25], Andrej Findor[26], Daniel Flichtentrei[27], Renata Franc[7], Biljana Gjoneska[28], June Gruber[29], Estrella Gualda[30,31], Yusaku Horiuchi[21], Toan Luu Duc Huynh[32], Agustin Ibanez[33,34], Mostak Ahamed Imran[35], Jacob Israelashvili[36], Katarzyna Jasko[37], Jaroslaw Kantorowicz[38], Elena Kantorowicz-Reznichenko[39], André Krouwel[40], Michael Laakasuo[41], Claus Lamm[42], Caroline Leygue[43], Ming-Jen Lin[44,45], Mohammad Sabbir Mansoor[46], Antoine Marie[19], Lewend Mayiwar[47], Honorata Mazepus[48,49], Cillian McHugh[50], John Paul Minda[51], Panagiotis Mitkidis[25,52], Andreas Olsson[53], Tobias Otterbring[54,55], Dominic J. Packer[56], Anat Perry[36], Michael Bang Petersen[19], Arathy Puthillam[57], Julián C. Riaño-Moreno[58,59], Tobias Rothmund[10], Hernando Santamaría-García[60], Petra C. Schmid[61], Drozdstoy Stoyanov[62], Shruti Tewari[63], Bojan Todosijević[64], Manos Tsakiris[65,66,67], Hans H. Tung[68,45], Radu G. Umbreş[69], Edmunds Vanags[70], Madalina Vlasceanu[71], Andrew Vonasch[72], Meltem Yucel[73,74], Yucheng Zhang[75], Mohcine Abad[76], Eli Adler[36], Narin Akrawi[77], Hamza Alaoui Mdarhri[76], Hanane Amara[78], David M. Amodio[1,79], Benedict G. Antazo[80], Matthew Apps[13], F. Ceren Ay[81,82], Mouhamadou Hady Ba[83], Sergio Barbosa[84,85], Brock Bastian[86], Anton Berg[41], Maria P. Bernal-Zárate[58], Michael Bernstein[87], Michał Białek[88], Ennio Bilancini[89], Natalia Bogatyreva[16], Leonardo Boncinelli[90], Jonathan E. Booth[91], Sylvie Borau[92], Ondrej Buchel[93,94], C. Daryl Cameron[95,96], Chrissie F. Carvalho[97], Tatiana Celadin[98], Chiara Cerami[99,100], Hom Nath Chalise[46], Xiaojun Cheng[101], Luca Cian[102], Kate Cockcroft[18], Jane Conway[103], Mateo Andres Córdoba-Delgado[60], Chiara Crespi[100,104], Marie Crouzevialle[61], Jo Cutler[12,13], Marzena Cypryańska[5], Justyna Dabrowska[105], Michael A. Daniels[106], Victoria H. Davis[22], Pamala N. Dayley[107], Sylvain Delouvee[108], Ognjan Denkovski[79], Guillaume Dezecache[109], Nathan A. Dhaliwal[106], Alelie B. Diato[110], Roberto Di Paolo[89], Marianna Drosinou[41], Uwe Dulleck[111,112,113,114], Jānis Ekmanis[70], Arhan S. Ertan[115], Tom W. Etienne[116], Hapsa Hossain Farhana[35], Fahima Farkhari[10], Harry Farmer[11], Ali Fenwick[117], Kristijan Fidanovski[118], Terry Flew[119], Shona Fraser[120], Raymond Boadi Frempong[121], Jonathan A. Fugelsang[122], Jessica Gale[72], E. Begoña Garcia-Navarro[30], Prasad Garladinne[63], Oussama Ghajjou[123], Theofilos Gkinopoulos[124], Kurt Gray[125], Siobhán M. Griffin[50], Bjarki Gronfeldt[2], Mert Gümren[126], Ranju Lama Gurung[46], Eran Halperin[36], Elizabeth Harris[1], Volo Herzon[41], Matej Hruška[26], Guanxiong Huang[127], Matthias F. C. Hudecek[128,129], Ozan Isler[111,112], Simon Jangard[53], Frederik J. Jørgensen[19], Frank Kachanoff[125], John Kahn[21], Apsara Katuwal Dangol[46], Oleksandra Keudel[130], Lina Koppel[131], Mika Koverola[41], Emily Kubin[132], Anton Kunnari[41], Yordan Kutiyski[116], Oscar Laguna[116], Josh Leota[133], Eva Lermer[129,134,135], Jonathan Levy[136,137], Neil Levy[8], Chunyun Li[91], Elizabeth U. Long[22], Chiara Longoni[138], Marina Maglić[7], Darragh McCashin[139], Alexander L. Metcalf[140], Igor Mikloušić[7], Soulaimane El Mimouni[78], Asako Miura[141], Juliana Molina-Paredes[60], César Monroy-Fonseca[142], Elena Morales-Marente[30], David Moreau[143], Rafał Muda[144], Annalisa Myer[74,145], Kyle Nash[133], Tarik Nesh-Nash[78], Jonas P. Nitschke[42], Matthew S. Nurse[146], Yohsuke Ohtsubo[147], Victoria Oldemburgo de Mello[22], Cathal O'Madagain[76], Michal Onderco[148], M. Soledad Palacios-Galvez[30], Jussi Palomäki[41], Yafeng Pan[53], Zsófia Papp[149], Philip Pärnamets[53], Mariola Paruzel-Czachura[150], Zoran Pavlović[151], César Payán-Gómez[152], Silva Perander[41], Michael Mark Pitman[18], Rajib Prasad[153], Joanna Pyrkosz-Pacyna[154], Steve Rathje[155], Ali Raza[156,157],

Gabriel G. Rêgo[158], Kasey Rhee[159], Claire E. Robertson[1], Iván Rodríguez-Pascual[30], Teemu Saikkonen[160], Octavio Salvador-Ginez[43], Waldir M. Sampaio[158], Gaia C. Santi[99], Natalia Santiago-Tovar[161], David Savage[162], Julian A. Scheffer[95], Philipp Schönegger[163,164], David T. Schultner[80], Enid M. Schutte[18], Andy Scott[133], Madhavi Sharma[46], Pujan Sharma[46], Ahmed Skali[165], David Stadelmann[121], Clara Alexandra Stafford[51,166,167], Dragan Stanojević[168], Anna Stefaniak[169], Anni Sternisko[1], Agustin Stoica[170], Kristina K. Stoyanova[171], Brent Strickland[76,172], Jukka Sundvall[41], Jeffrey P. Thomas[86], Gustav Tinghög[131], Benno Torgler[111,112,173], Iris J. Traast[80], Raffaele Tucciarelli[174,175], Michael Tyrala[176], Nick D. Ungson[177], Mete S. Uysal[178], Paul A. M. Van Lange[179], Jan-Willem van Prooijen[179], Dirk van Rooy[180], Daniel Västfjäll[181], Peter Verkoeijen[182], Joana B. Vieira[53], Christian von Sikorski[183], Alexander Cameron Walker[122], Jennifer Watermeyer[184], Erik Wetter[185], Ashley Whillans[186], Robin Willardt[61], Michael J. A. Wohl[169], Adrian Dominik Wójcik[187], Kaidi Wu[188], Yuki Yamada[189], Onurcan Yilmaz[190], Kumar Yogeeswaran[72], Carolin-Theresa Ziemer[10], Rolf A. Zwaan[182] & Paulo S. Boggio[158]

[1]Department of Psychology and Neural Science, New York University, New York, NY, USA. [2]School of Psychology, University of Kent, Canterbury, England. [3]Department of Economics, Middlesex University London, London, England. [4]Department of Strategy and Management, Norwegian School of Economics, Bergen, Norway. [5]SWPS University of Social Sciences and Humanities, Poznań, Poland. [6]Department of Psychological Sciences, College of William and Mary, Williamsburg, VA, USA. [7]Institute of Social Sciences Ivo Pilar, Zagreb, Croatia. [8]Department of Philosophy, Macquarie University, Sydney, NSW, Australia. [9]Stanford Graduate School of Business, Stanford University, Stanford, CA, USA. [10]Institute of Communication Science, Friedrich-Schiller University Jena, Jena, Germany. [11]School of Human Sciences, Institute for Lifecourse Development, University of Greenwich, London, England. [12]Department of Experimental Psychology, University of Oxford, Oxford, England. [13]Center for Human Brain Health, School of Psychology, University of Birmingham, Birmingham, England. [14]Department of Psychology, Macquarie University, Sydney, NSW, Australia. [15]KU Leuven, Leuven, Belgium. [16]National Research University Higher School of Economics (HSE), Moscow, Russia. [17]De La Salle University, Manila, Philippines. [18]Department of Psychology, University of the Witwatersrand, Johannesburg, South Africa. [19]Department of Political Science, Aarhus University, Aarhus, Denmark. [20]X University, Toronto, Canada. [21]Department of Government, Dartmouth College, Hanover, NH, USA. [22]Department of Psychology, University of Toronto, Toronto, ON, Canada. [23]Gatton College of Business and Economics, University of Kentucky, Lexington, KY, USA. [24]Department of Mass Communication, National University of Science and Technology (NUST), Islamabad, Pakistan. [25]Department of Management, Aarhus University, Aarhus, Denmark. [26]Faculty of Social and Economic Sciences, Comenius University, Bratislava, Slovakia. [27]IntraMed, Buenos Aires, Argentina. [28]Macedonian Academy of Sciences and Arts, North Macedonia, Republic of North Macedonia. [29]University of Colorado Boulder, Boulder, CO, USA. [30]ESEIS/COIDESO [ESEIS, Social Studies and Social Intervention Research Center; COIDESO, COIDESO, Center for Research in Contemporary Thought and Innovation for Social Development], University of Huelva, Huelva, Spain. [31]Faculty of Social Work, University of Huelva, Huelva, Spain. [32]WHU – Otto Beisheim School of Management, Vallendar, Germany. [33]Latin American Brain Health Institute (BrainLat), Adolfo Ibáñez University, Santiago, Chile. [34]Global Brain Health Institute, University of San Andrés, Buenos Aires, Argentina. [35]Department of Educational and Counselling Psychology, University of Dhaka, Dhaka, Bangladesh. [36]Psychology Department, The Hebrew University of Jerusalem, Jerusalem, Israel. [37]Institute of Psychology, Jagiellonian University, Kraków, Poland. [38]Institute of Security and Global Affairs, Leiden University, The Hague, Netherlands. [39]Erasmus School of Law, Erasmus University Rotterdam, Rotterdam, Netherlands. [40]Department of Political Science, Vrije University (VU) Amsterdam, Amsterdam, Netherlands. [41]Department of Digital Humanities, University of Helsinki, Helsinki, Finland. [42]Department of Cognition, Emotion, and Methods in Psychology, University of Vienna, Vienna, Austria. [43]School of Psychology, National Autonomous University of Mexico, Mexico City, Mexico. [44]Department of Economics, National Taiwan University, Taipei, Taiwan. [45]Center for Research in Econometric Theory and Applications, National Taiwan University, Taipei, Taiwan. [46]Tribhuvan University, Kirtipur, Nepal. [47]Department of Leadership and Organizational Behavior, BI Norwegian Business School, Oslo, Norway. [48]Institute of Security and Global Affairs, Leiden University, Leiden, Netherlands. [49]Faculty of Governance and Global Affairs, Leiden University, Leiden, Netherlands. [50]Department of Psychology, University of Limerick, Limerick, Ireland. [51]Department of Psychology, The University of Western Ontario, London, ON, Canada. [52]Center for Advanced Hindsight, Duke University, Durham, NC, USA. [53]Department of Clinical Neuroscience, Karolinska Institute, Solna, Sweden. [54]Department of Management, University of Agder, Kristiansand, Norway. [55]Institute of Retail Economics, Stockholm, Sweden. [56]Department of Psychology, Lehigh University, Bethlehem, PA, USA. [57]Department of Psychology, Monk Prayogshala, Mumbai, India. [58]Medicine Faculty, Cooperative University of Colombia, Villavicencio, Colombia. [59]Department of Bioethics, El Bosque University, Bogotá, Colombia. [60]Faculty of Medicine, Pontifical Javeriana University, Bogotá, Colombia. [61]Department of Management, Technology, and Economics, ETH Zürich, Zürich, Switzerland. [62]Department of Psychiatry and Medical Psychology, Research Institute, Medical University of Plovdiv, Plovdiv, Bulgaria. [63]Humanities and Social Sciences, Indian Institute of Management, Indore, India. [64]Institute of Social Sciences, Belgrade, Serbia. [65]Department of Psychology, Royal Holloway, University of London, London, England. [66]Center for the Politics of Feelings, School of Advanced Study, University of London, London, England. [67]Department of Behavioral and Cognitive Sciences, Faculty of Humanities, Education and Social Sciences, University of Luxembourg, Luxembourg City, Luxembourg. [68]Department of Political Science, National Taiwan University, Taipei, Taiwan. [69]Faculty of Political Science, National School for Political Studies and Public Administration, Bucharest, Romania. [70]Department of Psychology, University of Latvia, Riga, Latvia. [71]Department of Psychology, Princeton University, Princeton, NJ, USA. [72]Department of Psychology, Speech, and Hearing, University of Canterbury, Christchurch, New Zealand. [73]Department of Psychology and Neuroscience, Duke University, Durham, NC, USA. [74]Department of Psychology, University of Virginia, Charlottesville, VA, USA. [75]School of Economics and Management, Hebei University of Technology, Tianjin, PR China. [76]School of Collective Intelligence, Mohammed VI Polytechnic University, Ben Guerir, Morocco. [77]Institute for Research and Development-Kurdistan, Middle East, Iraq. [78]Impact For Development, North Africa, Morocco. [79]Department of Psychology, University of Amsterdam, Amsterdam, Netherlands. [80]Department of Psychology, Jose Rizal University, Mandaluyong, Philippines. [81]Department of Economics, Norwegian School of Economics, Bergen, Norway. [82]Telenor Research, Oslo, Norway. [83]Department of Philosophy, University Cheikh

Anta Diop, Dakar, Senegal. [84]School of Medicine and Health Sciences, University of Rosario, Bogotá, Colombia. [85]Moral Psychology and Decision Sciences Research Incubator, University of Rosario, Bogotá, Colombia. [86]School of Psychological Sciences, University of Melbourne, Parkville, VIC, Australia. [87]Department of Psychological and Social Sciences, Penn State Abington, Abington, PA, USA. [88]Institute of Psychology, University of Wrocław, Wrocław, Poland. [89]IMT School for Advanced Studies Lucca, Lucca, Italy. [90]Department of Economics and Management, University of Florence, Florence, Italy. [91]Department of Management, London School of Economics and Political Science, London, England. [92]Toulouse Business School, University of Toulouse, Toulouse, France. [93]Social Policy Institute of the Ministry of Labor, Family and Social Affairs of the Slovak Republic, Bratislava, Slovakia. [94]Department of Sociology, Tilburg University, Tilburg, Netherlands. [95]Department of Psychology, Penn State University, University Park, PA, USA. [96]Rock Ethics Institute, Penn State University, University Park, PA, USA. [97]Department of Psychology, Federal University of Santa Catarina, Florianópolis, Brazil. [98]Department of Economics, University of Bologna, Bologna, Italy. [99]IUSS Cognitive Neuroscience (ICoN) Center, Institute for Advanced Study of Pavia, Pavia, Italy. [100]Cognitive Computational Neuroscience Research Unit, Neurological Institute Foundation Casimiro Mondino, Pavia, Italy. [101]School of Psychology, Shenzhen University, Shenzhen, PR China. [102]Darden School of Business, University of Virginia, Charlottesville, VA, USA. [103]Institute for Advanced Study in Toulouse, Université Toulouse 1 Capitole, Toulouse, France. [104]Department of Brain and Behavioral Sciences, University of Pavia, Pavia, Italy. [105]Cracow University of Economics, Kraków, Poland. [106]UBC Sauder School of Business, University of British Columbia, Vancouver, BC, Canada. [107]Psychology Department, University of California - Los Angeles, Los Angeles, CA, USA. [108]Laboratory of Psychology: Cognition, Behavior, and Communication (LP3C), Rennes 2 University, Rennes, France. [109]Laboratory of Social and Cognitive Psychology, Clermont Auvergne University, CNRS, Clermont-Ferrand, France. [110]Cavite State University-General Trias City Campus, Cavite, Philippines. [111]School of Economics and Finance, Queensland University of Technology, Brisbane, QLD, Australia. [112]Center for Behavioural Economics, Society and Technology, Queensland University of Technology, Brisbane, QLD, Australia. [113]Crawford School of Public Policy, Australian National University, Canberra, ACT, Australia. [114]CESifo, University of Munich, Munich, Germany. [115]Department of International Trade, Boğaziçi University, Istanbul, Turkey. [116]Kieskompas - Election Compass, Amsterdam, Netherlands. [117]Hult International Business School Dubai, Dubai, UAE. [118]Department of Social Policy and Intervention, University of Oxford, Oxford, England. [119]Department of Media and Communications, University of Sydney, Sydney, NSW, Australia. [120]Department of Psychiatry, University of the Witwatersrand, Johannesburg, South Africa. [121]University of Bayreuth, Bayreuth, Germany. [122]Department of Psychology, University of Waterloo, Waterloo, ON, Canada. [123]Department of Peace Studies, University of Bradford, Bradford, England. [124]Philosophy and Social Studies Department, Rethymno, Greece. [125]Department of Psychology and Neuroscience, University of North Carolina at Chapel Hill, Chapel Hill, NC, USA. [126]Department of Economics, Koc University, Istanbul, Turkey. [127]Department of Media and Communication, City University of Hong Kong, Kowloon Tong, Hong Kong. [128]University of Regensburg, Regensburg, Germany. [129]FOM University of Applied Sciences, Essen, Germany. [130]Graduate School for Transnational Studies, Free University of Berlin, Berlin, Germany. [131]Department of Management and Engineering, Linköping University, Linköping, Sweden. [132]Department of Psychology, University of Koblenz-Landau, Landau, Germany. [133]Department of Psychology, University of Alberta, Edmonton, Canada. [134]LMU Center for Leadership and People Management, Ludwig Maximilian University of Munich, Munich, Germany. [135]Ansbach University for Applied Sciences, Ansbach, Germany. [136]Baruch Ivcher School of Psychology, Interdisciplinary Center Herzliya (IDC), Herzliya, Israel. [137]Department of Neuroscience and Biomedical Engineering, Aalto University, Espoo, Finland. [138]Questrom School of Business, Boston University, Boston, MA, USA. [139]School of Psychology, Dublin City University, Dublin, Ireland. [140]University of Montana, Missoula, MT, USA. [141]Graduate School of Human Sciences Human Sciences, Osaka University, Suita, Japan. [142]SEELE Neuroscience, Mexico City, Mexico. [143]School of Psychology, University of Auckland, Auckland, New Zealand. [144]Faculty of Economics, Maria Curie-Skłodowska University, Lublin, Poland. [145]Department of Psychology, The City University of New York (CUNY) Graduate Center, New York, NY, USA. [146]Australian National Centre for the Public Awareness of Science, Australian National University, Canberra, ACT, Australia. [147]Department of Social Psychology, Graduate School of Humanities and Sociology, University of Tokyo, Tokyo, Japan. [148]Department of Public Administration and Sociology, Erasmus University Rotterdam, Rotterdam, Netherlands. [149]Center for Social Sciences, Hungarian Academy of Sciences Center of Excellence, Budapest, Hungary. [150]Institute of Psychology, University of Silesia, Katowice, Poland. [151]Department of Psychology, University of Belgrade, Belgrade, Serbia. [152]Department of Biology, Faculty of Natural Sciences, Universidad del Rosario, Bogotá, Colombia. [153]Vidyasagar College For Women, Kolkata, India. [154]AGH University of Science and Technology, Kraków, Poland. [155]Department of Psychology, University of Cambridge, Cambridge, England. [156]Department of Computer Science, University of Colorado Boulder, Boulder, CO, USA. [157]Institute of Cognitive Science, University of Colorado Boulder, Boulder, CO, USA. [158]Social and Cognitive Neuroscience Laboratory, Mackenzie Presbyterian University, São Paulo, Brazil. [159]Stanford University, Stanford, CA, USA. [160]Department of Biology, University of Turku, Turku, Finland. [161]Cooperative University of Colombia, Bogotá, Colombia. [162]Newcastle Business School, University of Newcastle, Callaghan, NSW, Australia. [163]Department of Philosophy, University of St Andrews, St Andrews, Scotland. [164]School of Economics and Finance, University of St Andrews, St Andrews, Scotland. [165]Department of Global Economics and Management, University of Groningen, Groningen, Netherlands. [166]Brain and Mind Institute, University of Western Ontario, London, ON, Canada. [167]Western Interdisciplinary Research Building, University of Western Ontario, London, ON, Canada. [168]Department of Sociology, University of Belgrade, Belgrade, Serbia. [169]Department of Psychology, Carleton University, Ottawa, ON, Canada. [170]National University of Political Studies and Public Administration (SNSPA), Bucharest, Romania. [171]Research Institute at Medical University of Plovdiv), Division of Translational Neuroscience, Plovdiv, Bulgaria. [172]Department of Cognitive Science, ENS, EHESS, CNRS, Institut Jean Nicod, PSL Research University, Paris, France. [173]CREMA - Center for Research in Economics, Management and the Arts, Basel, Switzerland. [174]The Warburg Institute, School of Advanced Study, University of London, London, England. [175]Institute of Cognitive Neuroscience, University College London, London, England. [176]Institute for Emerging Market Studies, The Hong Kong University of Science and Technology, Kowloon, Hong Kong. [177]Department of Psychology, Susquehanna University, Selinsgrove, PA, USA. [178]Psychology Department, Dokuz Eylül University, İzmir, Turkey. [179]Department of Experimental and Applied Psychology, VU Amsterdam, Amsterdam, Netherlands. [180]Research School of Psychology, Australian National University, Canberra, ACT, Australia. [181]Department of Behavioural Sciences and Learning (IBL), Linköping University, Linköping, Sweden. [182]Department of Psychology, Education and Child Studies, Erasmus University Rotterdam, Rotterdam, Netherlands. [183]University of Koblenz-Landau, Landau, Germany. [184]Health Communication Research Unit, School of Human and Community Development, University of the Witwatersrand, Johannesburg, South Africa. [185]Department of Business Administration, Stockholm School of Economics, Stockholm, Sweden. [186]Harvard Business School, Harvard University, Cambridge, MA, USA. [187]Nicolaus Copernicus University, Toruń, Poland. [188]University of California, San Diego, La Jolla, CA, USA. [189]Kyushu University, Fukuoka, Japan. [190]Department of Psychology, Kadir Has University, Istanbul, Turkey. ✉email: jay.vanbavel@nyu.edu; flavio.azevedo@uni-jena.de

