## [Peer Review File · Nature Communications]

National identity predicts public health support during a global pandemicEditorial Note: This manuscript has been previously reviewed at another journal that is not operating a transparent peer review scheme. This document only contains reviewer comments and rebuttal letters for versions considered at *Nature Communications*.

Reviewers' comments:

Reviewer #2 (Remarks to the Author):

I have considered the authors' comments, and re-read the main body of their article. As you can see from my two comments, below, I am less sure about the robustness of the analysis than the other reviewers. But I have no objections to the publication of the revised paper. My concerns are with the conception and design of the exercise, and given the other reviewers do not share these concerns, I don't think they should hold up the article's publication.

For the record, my thoughts on the authors' comments and paper revisions are as follows:

My original point was that social solidarity/cohesion is likely to be associated with compliant behaviour among citizens. National identity is one, particular, manifestation of this 'we-feeling'. Maybe unsurprisingly, the authors find a positive correlation between national identity and compliance. They use this finding to suggest a positive benefit of stoking/enhancing feelings of national identity. My doubt about this would simply be that other forms of social identity are also likely to be important (as the authors acknowledge). Yet unless these other forms – social group identity, social trust etc – are explicitly measured and included in the model, it is difficult to identify the particular role played by national identity. I'm simply not convinced that finding a positive effect on compliance of one form of social identity/solidarity significantly advances social science understanding (or indeed of public policy). In their response, the authors do not mention testing the role of different forms of social identity (beyond partisanship). I suppose this means no such measures were included in their survey. If so, fine. But as a result, the results are less impressive and useful than I think they might have been.

I think these weaknesses are also reflected in the research design, which essentially deploys a single method for identifying the association between national identity and compliance. I think this shortcoming is less serious than the absence of measures of social identity/solidarity (above). But it could have been addressed more clearly by leveraging aggregate data alongside individual-level data.

Reviewer #3 (Remarks to the Author):

I previously reviewed an earlier version of this paper and made several comments and criticisms. The revisions the authors have undertaken have largely satisfied my concerns. If I were doing this work myself I would have opted for a different measure of national identity, but these are matters upon which reasonable people can disagree and is in the realm of scholarly debate. I am happy to see the paper published as is.

Reviewer #4 (Remarks to the Author):

The authors have done a very good job revising their manuscript. They do a better job distinguishing national identity and national narcissism, do not make questionable claims, and do a better job connecting their research to possible policy efforts. I also appreciate that they analyzed the impact of type of sampling and included the new Figure 3. This research is an important contribution to the literature.

I have a couple of quibbles, one in the paper and one in the authors' response. On page 7 of the paper, the authors write: "National narcissism then predicts greater preoccupation with maintaining a positive image of the nation than with the well-being of fellow citizens.... Thus, national narcissists may be less inclined to engage in behaviors to prevent the spread of COVID-19...." I get what the authors are saying, but I'm not convinced. Why wouldn't a person who is a national narcissist, and therefore desirous of a positive national image, be more likely to do what is necessary to dampen COVID-19? It makes a country look bad to have high COVID rates and to look good when rates are low (simply compare the U.S. or Italy to New Zealand, or Sweden and Norway). If having higher COVID numbers leads to more negative press and a lower international reputation, it would make sense to think that national narcissists would want their country's numbers to be low.

In the comments from the authors (and thank you for having your comments be clear and responsive to reviewer concerns), I'm not convinced by the argument on page 12. The authors write, "since country is not the unit of analysis, and there is no underlying need to have a sample of 500 people per each country (e.g., the vast majority of studies in psychology have less than 500 people across all conditions), we prefer to include the complete sample in the main paper." I think this is disingenuous. Most psychology studies are experiments with a small number of conditions. The important consideration is that participants are randomly assigned to condition, so there just need to be enough cases to test the impact of the conditions. Survey research is very different and depends on a random sample of respondents. A much larger sample size is needed in survey research than in experiments to be able to test the effects of survey responses across multiple variables. I still think the very small sample sizes in some of the countries is a problem, but the information now included in the revised manuscript is clearer and the claims made are not overly broad.

Reviewer #5 (Remarks to the Author):

I have been added as a new reviewer to this paper. I am divided about its contribution. On the one hand, the presented cross-national study has clear strengths. It collected data from a large range of countries and aimed to address a question of high relevance to the situation we are facing. On the other hand, there are many weaknesses and limitations, especially in terms of the data and methods. Some may be addressed through re-analyses and use of new, publicly available data as I outline below.

First, most of the datasets used are not representative. As a result, only 20 out of 65 countries can be meaningfully compared. The authors write,

"Although we managed to collect data from 67 countries and territories, we were nevertheless unable to ensure representative samples from many countries or even conduct our survey in other countries (especially in many African countries as well as the Middle East). Therefore, our inferences apply to nations where we managed to complete this research and the specific samples we obtained."

However, in my understanding, inferences based on data from 67 countries (with most being non-representative samples) cannot be used to make inferences about the 20 countries that did have representative data (for that, only these 20 countries would need to be analysed. I understand that moderations were ran, which may attenuate this critique, but if so it could be more clearly conveyed when and where it is reported).

Moreover, I was surprised that the analyses were conducted without a formal test of measurement invariance, which is long-established standard in cross-national research. That is, the authors need to show that the factor structures of their measures show at least metric or (better) scalar invariance. This is needed for any analyses that makes use of data from more than one country (regardless of controlling for the multi-level structure). One reason for why I think it will be difficult to establish measurement invariance is that reliabilities for the measures vary a lot between the countries. I didn't seem to find any information about it in the main document, so I calculated them myself. If I did that correctly, many countries cannot be used in analyses involving certain variables as their reliability

simple is too low.

For instance, for the hygiene measure, 9 countries have unacceptable reliabilities and would need to be excluded:

CR -0,021
HN 0,092
CU 0,297
PA 0,351
NI 0,367
GT 0,423
SV 0,493
EC 0,510
BO 0,580

If I removed the correct item for the spatial distancing variable, 14 countries have unacceptable reliabilities for this measure and need to be excluded from respective analyses:

AR 0,553
BO -0,424
CR -0,152
HN 0,219
PA 0,234
UY 0,315
VE 0,378
PY 0,434
CL 0,434
GT 0,441
DO 0,492
SV 0,501
EC 0,556
CN 0,579

For policy support, things look better with only 4 countries having an unreliable measurement

HN 0,122
GT 0,423
PE 0,503
DO 0,584

And only two countries for collect. narcissism:

CN 0,498
HN 0,572
GH 0,585

It is important to note that it is very lenient to take .6 as a reliability cutoff. .65 would be more normal for multi-item scales and that would lead to the exclusion of more countries. Thus, this all raises the question of whether the measures indeed measured the same thing in each country. One way the authors could and should address this is to conduct measurement invariance test and see whether dropping further items would allow them to establish invariance as well as more acceptable reliabilities. I provide some references here that outline ways of doing this when country number is high.

<https://www.tandfonline.com/doi/full/10.1080/10705511.2017.1304822>

<https://www.frontiersin.org/articles/10.3389/fpsyg.2019.01507/full>

Because most of the data is non-representative, any graphs that suggest that the countries can be compared should be removed. For instance, Figure 3 sorts the countries by strength of association. A reader may easily jump to the conclusion that these coefficients can be compared. The graph could be moved to the SOM and should also include confidence intervals. Alternatively, it could focus on the

20 representative countries, but, again, this would pre-require that measurement invariance has been established.

I am somewhat uncertain about the contribution of the paper given the reliance on correlational and self-report data. It may be little surprising that social attitudes are weakly correlated with self-reported behaviours within the same national context. However, with new data that is publicly available, the authors could convince me about the robustness and behavioral implications of their findings. Many large-scale survey programs with representative samples measure national identification. For instance, the newest wave of the World Values Survey that just came out measures national identity with items such as how close people feel to their country. This representative data is available for many countries (including a variety of non-WEIRD) and could be used to robustly test whether there is an association between national identity and actual health behaviour such as social distancing. The latter behaviour can be measured for instance through Google Mobility data <https://www.google.com/covid19/mobility/> that is available at many time points. Such a demonstration would go a long way. (By the way, similar tests could also be conducted with the 20 countries at the aggregate country level, although power would likely be low).

The title and various parts of the paper use language that implies causality (“predicts”), but what we are dealing with are associations (as the authors acknowledge). I understand that “predicts” can be used in an analytical sense (i.e., as in x predicts y in a regression model), but the paper should avoid such language throughout and especially in the title and abstract to not lead readers to false conclusions.

The study was not preregistered and I was somewhat confused by the role of collective narcissism in this study. It wasn't entirely clear for me why the variable was of interest. The authors surprisingly find that HDI was related to less public health support. They write,

“In other words, citizens in countries with higher scores on the global Human Development Index also reported less support for two of our COVID-19 public health measures. However, we should note that our dataset includes data from very few African countries, many of which have relatively low HDI scores but seem thus far to have fared better in the pandemic than higher-HDI countries.”

Could the authors present a formal test of whether African countries indeed were outliers that drove the effect?

Reviewers' comments:

Reviewer #2 (Remarks to the Author): While we were grateful that Reviewer 2 had “no objections to the publication of the revised paper”, they made the following comments and we have responded to each of them in our revision.

2.1. “My original point was that social solidarity/cohesion is likely to be associated with compliant behaviour among citizens. National identity is one, particular, manifestation of this ‘we-feeling’. Maybe unsurprisingly, the authors find a positive correlation between national identity and compliance. They use this finding to suggest a positive benefit of stoking/enhancing feelings of national identity. My doubt about this would simply be that other forms of social identity are also likely to be important (as the authors acknowledge). Yet unless these other forms – social group identity, social trust etc – are explicitly measured and included in the model, it is difficult to identify the particular role played by national identity. I’m simply not convinced that finding a positive effect on compliance of one form of social identity/solidarity significantly advances social science understanding (or indeed of public policy). In their response, the authors do not mention testing the role of different forms of social identity (beyond partisanship). I suppose this means no such measures were included in their survey. If so, fine. But as a result, the results are less impressive and useful than I think they might have been.”

Thank you for this thoughtful comment. We agree that other social identities may be important and, unfortunately, our survey did not include measures of identification with social groups other than the nation. However, we did include collective narcissism--which is a form of identity--and the effects of national identification were not only consistent after adjusting for collective narcissism, but also statistically stronger. We have now added these new analyses to the paper showing that national identification is a stronger predictor than this other form of identity:

*“We conducted tests comparing the size of these coefficients and found that for all public health measures, the coefficients for national identification were stronger than the coefficients for national narcissism and political ideology. Taken together, the three predictors accounted for 8% of the person-level variance of the contact measure, for 8% of the person-level variance of the hygiene measure, and 5% of the person-level variance of the policy support measure. The coefficients for individual countries are displayed in **Figure 2** and **Figure 3**.”*

We want to point out that social identity is not a cure all. For example, in the United States, identification with the Republican party is one of the key predictors of failing

to follow the COVID19 public health measures. We have now mentioned this issue in the general discussion section as we agree that it is worthy of future investigation:

“There is reason to believe that other forms of group identification can undercut public health. For instance, partisanship within countries (i.e., when people strongly identify with a specific political party) is associated with risky behavior (Alcott et al., 2020; Gadarian et al., 2020; Gollwitzer et al., 2020). For example, one study that used geo-tracking data from 15 million smartphones in the US found that counties that voted for a Republican (Donald Trump) over a Democrat (Hillary Clinton) exhibited 14% less spatial distancing during the early stages of the pandemic (Gollwitzer et al., 2020). These partisan gaps in distancing predicted subsequent increases in infections and mortality in counties that voted for Donald Trump. Moreover, partisanship was a stronger predictor of distancing than many other economic or social factors (e.g., county-level income, population density, religion, age, and state policy). This may be due to leadership, social norms, and media consumed by people from different identity groups. As such, stronger group identification is not always associated with engagement in public-healthy behavior.”

2.2. “I think these weaknesses are also reflected in the research design, which essentially deploys a single method for identifying the association between national identity and compliance. I think this shortcoming is less serious than the absence of measures of social identity/solidarity (above). But it could have been addressed more clearly by leveraging aggregate data alongside individual-level data.”

Thank you for raising this issue. We agree that our paper would be strengthened by including additional data using a complementary method and we have now added a second study with new aggregate data from around the world to address this concern. The good news is that the results replicate using this approach! Please see our detailed response to reviewer 5 (below).

Reviewer #3 (Remarks to the Author):

3.1 Reviewer 3 wrote: “I previously reviewed an earlier version of this paper and made several comments and criticisms. The revisions the authors have undertaken have largely satisfied my concerns. If I were doing this work myself I would have opted for a different measure of national identity, but these are matters upon which reasonable people can

disagree and is in the realm of scholarly debate. I am happy to see the paper published as is.”

We are grateful for this positive assessment of our revised manuscript and would like to thank the Reviewer for all their helpful comments on the previous draft.

Reviewer #4 (Remarks to the Author):

4.1 Reviewer 4 also thought we “have done a very good job revising their manuscript”. They said: “They do a better job distinguishing national identity and national narcissism, do not make questionable claims, and do a better job connecting their research to possible policy efforts. I also appreciate that they analyzed the impact of type of sampling and included the new Figure 3. This research is an important contribution to the literature.”

We are grateful to the Reviewer for these positive comments, and we seek to address the two outstanding comments below.

4.2 “ On page 7 of the paper, the authors write: “National narcissism then predicts greater preoccupation with maintaining a positive image of the nation than with the well-being of fellow citizens....Thus, national narcissists may be less inclined to engage in behaviors to prevent the spread of COVID-19....” I get what the authors are saying, but I’m not convinced. Why wouldn’t a person who is a national narcissist, and therefore desirous of a positive national image, be more likely to do what is necessary to dampen COVID-19? It makes a country look bad to have high COVID rates and to look good when rates are low (simply compare the U.S. or Italy to New Zealand, or Sweden and Norway). If having higher COVID numbers leads to more negative press and a lower international reputation, it would make sense to think that national narcissists would want their country’s numbers to be low.”

Thank you for the opportunity to clarify our reasoning. Reviewer 4 is right that one would expect those scoring high in national narcissism to protect a country’s image. However, we believe many of them are more interested in making the country look good, rather than in actually engaging in health-protective behaviour. This is because national narcissism seems to predict greater investment in short-term, image-enhancing strategies rather than long-term, substantive public-health outcomes. This short term approach often leads to long term failures (which is the opposite of the approach taken by New Zealand).

For example, in a recent unpublished study conducted by our co-authors last year, national narcissism in the US predicted support for conducting less COVID-19 testing in order to make the US infection rates look better. Similarly, a recent paper focusing on the relationship between national narcissism and environmental policies finds that national narcissism predicts greater support for engaging in promoting the nation's image as pro-environmental, but lower support for actual pro-environmental policies (Cislak et al., 2021; Journal of Environmental Psychology). Thus, those scoring high in national narcissism likely want their country's numbers to "look low", more so than "be low". We now edited the relevant section on p. X read:

"National narcissism then predicts greater preoccupation with maintaining a positive image of the nation than with the well-being of fellow citizens (Cislak et al., 2018; Marchlewska et al., 2020). Thus, in a crisis, national narcissists may prefer to invest in short-term image enhancement than in long-term solutions (see also Cislak et al., 2021). They may then be less inclined to engage in behaviors to prevent the spread of COVID-19--or even acknowledge the risks associated with the pandemic in their home country (Nowak et al., 2020)."

4.2 "In the comments from the authors (and thank you for having your comments be clear and responsive to reviewer concerns), I'm not convinced by the argument on page 12. The authors write, "since country is not the unit of analysis, and there is no underlying need to have a sample of 500 people per each country (e.g., the vast majority of studies in psychology have less than 500 people across all conditions), we prefer to include the complete sample in the main paper." I think this is disingenuous. Most psychology studies are experiments with a small number of conditions. The important consideration is that participants are randomly assigned to condition, so there just need to be enough cases to test the impact of the conditions. Survey research is very different and depends on a random sample of respondents. A much larger sample size is needed in survey research than in experiments to be able to test the effects of survey responses across multiple variables. I still think the very small sample sizes in some of the countries is a problem, but the information now included in the revised manuscript is clearer and the claims made are not overly broad."

We agree that experimental and survey studies have different sampling considerations. However, in multilevel analyses, such as ours, the biggest concern is often the higher-level sample size (Hox & Maas, 2004). Retaining more samples allowed us to include a large enough number of countries for multilevel analysis. However, this could have implications for the precision of the coefficients in each nation so we have noted this in the supplement.

Reviewer #5 (Remarks to the Author):

5.1 Reviewer 5 commented, “First, most of the datasets used are not representative. As a result, only 20 out of 65 countries can be meaningfully compared. The authors write,

“Although we managed to collect data from 67 countries and territories, we were nevertheless unable to ensure representative samples from many countries or even conduct our survey in other countries (especially in many African countries as well as the Middle East). Therefore, our inferences apply to nations where we managed to complete this research and the specific samples we obtained.”

However, in my understanding, inferences based on data from 67 countries (with most being non-representative samples) cannot be used to make inferences about the 20 countries that did have representative data (for that, only these 20 countries would need to be analysed. I understand that moderations were ran, which may attenuate this critique, but if so it could be more clearly conveyed when and where it is reported).”

Thank you for your comments. Although we agree that our conclusions cannot easily generalize to all citizens in all these countries, we disagree that our samples constitute a limitation for a couple of reasons. First, the overwhelming majority of published papers in the field of psychology have no representative samples (e.g., we guess that less than 1% have such samples and the vast majority are undergraduate or MTURK samples from WEIRD countries). Second, including all samples allowed us to achieve a sample size required (> 50) for multilevel analyses, where higher-level (in our case, country-level) is usually a concern (Hox & Maas, 2004). Moreover, multilevel models take into account different numbers of observations within groups.

We aim to provide an open acknowledgment of limitations in regards to non-representative samples and have quantitatively assessed the role of national identification in both representative and non-representative samples. The good news is that there is really no difference in the main conclusions depending on the samples we analyze. If anything, there seems to be a stronger relationship in the most representative samples. We have now included more details of this analysis in the paper and think it helps strengthen the overall conclusions.

We have now included a brief note in the paper and a more detailed analysis in the supplement noting that the representativeness of the samples has no bearing on the conclusions. Specifically, we find almost identical patterns of results (with identification predicting health measures) in both representative samples and non-

representative samples. As such, we have strong empirical grounds to believe this issue does not impact any of our conclusions:

'We collected data in 67 countries. In 28 of these countries, we were able to obtain representative samples in terms of sex, age, and education. We collected convenience samples in 36 countries, and in three countries the sampling was mixed. We examined possible differences between countries as function of the representativeness of the sample by including a contrast coded variable (1 = representative, 0 – mixed, -1 = non-representative) at the country-level in the models that examined relationships between our three public health measures (spatial distancing, physical hygiene, and policy support) and our three predictor variables (national narcissism, national identification, and political ideology).

We found that means for the three public health measures were higher in non-representative samples than they were in representative samples. We found one significant moderating effect for slopes ($g_{021} = .038$, $t = 2.55$, $p = .014$). In the analysis of spatial distancing, the slope for national identification was weaker for countries that had obtained non-representative samples than it was for countries that were able to obtain representative samples. Although it was slightly weaker, we should note that the slope for countries with non-representative samples (.08) was still significantly different from 0 ($p < .001$)."

5.2 "Moreover, I was surprised that the analyses were conducted without a formal test of measurement invariance, which is long-established standard in cross-national research. That is, the authors need to show that the factor structures of their measures show at least metric or (better) scalar invariance. This is needed for any analyses that makes use of data from more than one country (regardless of controlling for the multi-level structure). One reason for why I think it will be difficult to establish measurement invariance is that reliabilities for the measures vary a lot between the countries. I didn't seem to find any information about it in the main document, so I calculated them myself. If I did that correctly, many countries cannot be used in analyses involving certain variables as their reliability simple is too low." Reviewer 5 proceeds with examples of countries with unacceptable reliabilities,

Reviewer 5 further wrote: "Thus, this all raises the question of whether the measures indeed measured the same thing in each country. One way the authors could and should address this is to conduct measurement invariance test and see whether dropping further items would allow them to establish invariance as well as more acceptable reliabilities."

Thank you for this comment. As you note, it is very difficult--if not impossible--to establish measure invariance across this many countries. We were less concerned by invariance testing given recent findings showing that non-invariance might be less problematic for cross-cultural research than initially assumed (e.g., Welzel et al., 2021; Sociological Methods & Research). However, we do agree with the Reviewer that it is important to consider the reliability of our measures and we employed a specific analytic approach to address this issue.

*Specifically, we presented a three-level analysis (items nested with person, persons nested with countries). Such analyses take into account **individual level** reliabilities in responses. This is a step further than country-level reliabilities, which can mask the likelihood that a measure is more or less reliable across individuals within a country. One of our co-authors, who is a pioneer in multi-level modeling (see papers by Nezlek, 2001; 2008), suggested that this was the most powerful approach to address this issue. Indeed, this is one of the major advantages of using a multi-level model as an analytic approach. We also accounted for Bayesian shrinkage in our models to downweight less reliable observations. We sought to explain our approach more clearly in the manuscript:*

“We analyzed the data using multi-level models in which persons were treated as nested within countries (Nezlek, 2010). We also included a measurement level to control for individual differences in how consistently people responded to items that were meant to measure the same construct. Our analyses estimated relationships at the individual level while controlling for country-level differences. For example, did people who had a stronger national identification endorse public health measures such as social distancing more strongly than people with a weaker national identification? A set of regression coefficients was estimated for each country, and the means of these coefficients were tested for statistical significance. Moreover, the standard errors of these coefficients incorporated “Bayesian shrinkage” meaning that less reliable observations (countries and individuals) influenced parameter estimates less than more reliable observations.”

The reviewer also reported countries with a reliability $<.60$ and raised concerns about using these countries in the analyses. For thoroughness, we decided to directly test whether this was an issue by comparing all countries with reliabilities $<.60$ on our key independent and dependent measures to those without reliability issues. Thankfully, the main findings of the relation between national identification and each of our public health measures were not moderated by samples with low reliability measures. In short, reliability did not account for any of our findings and does not qualify any of our main conclusions. Thanks for encouraging us to

consider and test this possibility. We are now more confident in our main conclusions. As we note in the supplement:

“Although our analyses took into account individual differences in the reliability of our outcomes and the unreliability of slopes, to examine more thoroughly if unreliability may have confounded our results, we conducted a series of analyses in which we examined if the coefficients (slopes and intercept) varied as a function of whether the measures in an analysis were reliable or not. Following the guidelines suggested by Shrout (1998), we defined reliable as .6 or above.

For the analyses of spatial distancing, the outcome and predictors were reliable for 50 countries, for physical hygiene, they were reliable for 53 countries, and for policy support they were reliable for 57 countries. Similar to how we examined the possible influence of the representativeness of the sample, we added a contrast coded variable (1 = reliable, -1 = not reliable) at the country-level in the models that examined relationships between our three outcomes and three predictors. These analyses found no significant effects for the reliability of our measures for intercepts or slopes.”

*We would also like to stress that significant results would be obtained **despite** unreliability, not because of it. Reliability is a form of random error. The more error, the less likely one is to find a relationship. This would make the effect size estimates from nations with lower reliability highly conservative. Furthermore, because all of our scales are single factors with very few items, and because we focus on associations between variables rather than estimations of mean levels of different constructs, individual level reliability is a bigger concern than invariance. We are happy to add this to the manuscript if you think it would be useful for future readers.*

5.3 Because most of the data is non-representative, any graphs that suggest that the countries can be compared should be removed. For instance, Figure 3 sorts the countries by strength of association. A reader may easily jump to the conclusion that these coefficients can be compared. The graph could be moved to the SOM and should also include confidence intervals. Alternatively, it could focus on the 20 representative countries, but, again, this would pre-require that measurement invariance has been established.

We agree that this is a general issue for any figure that includes multiple groups with measurement effort differences. Moreover, as we noted above, we find no differences between the countries with representative samples and those without.

We have further clarified this in the manuscript and supplement and are happy to expand on these analyses in the supplement if you think it would be helpful for readers.

It is also worth noting that the coefficients are in the same direction for virtually every measure for every country we studied and we make no major comparisons or conclusions about differences between specific countries. As such, we prefer to include the figures for illustrative purposes about the general consistency of the findings (which is consistent with the language in our paper). We are happy to include an acknowledgment of the differences in error in the figure caption, if you think it would help. But again, we are reluctant to make the differences salient when that is not the thrust of the paper. Instead, we have tried to highlight the general pattern reflected in the figure.

We have also created new figures that include 95% confidence intervals for every country and coefficient. These are quite hard to read and take up a lot of space, so we have placed them in the supplement for people who are extremely interested in these details. But I have pasted them here for you to see if you would like us to move them into the body of the paper:

Figure S1. Relation between national identification and public health measures in 67 countries and territories. The coefficients reflecting the relation between national identity and each of the health measures are presented for each country. The relation with physical contact (left), hygiene (center), and policy support (right) are presented with 95% confidence intervals.

Figure S2. Relation between national narcissism and public health measures in 67 countries and territories. The coefficients reflecting the relation between national narcissism and each of the health measures are presented for each country. The relation with physical contact (left), hygiene (center), and policy support (right) are presented with 95% confidence intervals.

Figure S3. Relation between political ideology and public health measures in 67 countries and territories. The coefficients reflecting the relation between political ideology and each of the health measures are presented for each country. The relation with physical contact (left), hygiene (center), and policy support (right) are presented with 95% confidence intervals.

5.4 I am somewhat uncertain about the contribution of the paper given the reliance on correlational and self-report data. It may be little surprising that social attitudes are weakly correlated with self-reported behaviours within the same national context. However, with new data that is publicly available, the authors could convince me about the robustness and behavioral implications of their findings. Many large-scale survey programs with representative samples measure national identification. For instance, the newest wave of the World Values Survey that just came out measures national identity with items such as how close people feel to their country. This representative data is available for many countries (including a variety of non-WEIRD) and could be used to robustly test whether there is an association between national identity and actual health behaviour such as social distancing. The latter behaviour can be measured for instance through Google Mobility data

<https://www.google.com/covid19/mobility/> that is available at many time points. Such a demonstration would go a long way. (By the way, similar tests could also be conducted with the 20 countries at the aggregate country level, although power would likely be to low).

We are extremely grateful to the reviewer for making this suggestion. We followed this recommendation and conducted a second study using Google Community Mobility Reports and correlated physical distancing with an index of national identification from the World Value Survey (combining items measuring national pride and closeness to their own country). We were able to obtain data from 42 countries which had available data for both the national identification and the mobility scores.

As you can see in the paper, these results clearly replicate the findings from Study 1 using aggregate indices of physical movement and national identification. We note in the paper:

*“Study 1 relied on self-report measures. To test the robustness of our predictions, we sought to conceptually replicate our findings using publicly available indices of national identity as well as actual behavior change during the pandemic. To this end, we relied on two publicly available datasets: the World Values Survey (Haerpfer et al., 2020) and the COVID-19 Google Community Mobility Reports which indicate how people’s physical movement has changed in response to COVID-19 policies (available at www.google.com/covid19/mobility/). We created an index of national identification using the two relevant items from the World Value Survey (i.e., national pride and closeness to their nation) and an index of physical mobility by averaging community movement across all available places (i.e., retail and recreation, groceries and pharmacies, parks, transit stations, workplaces, and residential). See **Methods** for details about the sample and measures.*

We examined whether countries with higher average national identification would also show stronger change in mobility in response to COVID-19 restrictions during April and May 2020--the period in which we collected most of the samples in Study 1. We conducted our analysis for the full sample of 42 countries in which aggregate data which was publically available for both for the national identification and the mobility scores.

Replicating the pattern of results from Study 1, national identification was positively associated with reduced spatial mobility, $r = .40$, $p = .008^1$. The observed

¹ Please see the Supplement for separate correlations for each of the places and the two indices of national identifications.

association at the aggregate level was moderate to strong. Thus, we found evidence both at the person-level and country-level showing a link between national identification and support for and engagement with public health behaviors.”

As these new analyses provide clear converging evidence for the self-report data in 67 countries, we can be even more confident in the results of the paper. If the editor believes that the paper would be strengthened by swapping the order of studies, please let us know as we can see merits to both approaches.

5.5 “The title and various parts of the paper use language that implies causality (“predicts”), but what we are dealing with are associations (as the authors acknowledge). I understand that “predicts” can be used in an analytical sense (i.e., as in x predicts y in a regression model), but the paper should avoid such language throughout and especially in the title and abstract to not lead readers to false conclusions”.

Thanks for raising this issue. Although we tried to be careful with our language, we realize it can be interpreted differently by various audiences. As such, we have gone through the paper again to mitigate any confusion about this issue. We are happy to change any additional instances that could be confusing to readers.

5.6 “The study was not pre-registered and I was somewhat confused by the role of collective narcissism in this study. It wasn’t entirely clear for me why the variable was of interest”.

As Reviewer 5 notes, the study was not pre-registered due to the urgency of conducting the research around the world during the pandemic. However, we planned to test the effect of collective narcissism alongside the effect of national identification at the stage of survey design. Just as it is important to distinguish self-esteem from narcissism, it is important to distinguish genuine national identification, that is the importance of one's identity to the self and satisfaction with group membership, from the more defensive collective narcissism, which captures beliefs about ingroup superiority and entitlement to special treatment. This also allowed us to distinguish national identification from other forms of identity (as we noted in our response to one of the other reviewers, above). We have tried to make the conceptual argument more transparent in our revision. As we explain in the manuscript:

“National identification tends to correlate positively with national narcissism because they both assume a positive evaluation of one’s nation. However, they predict different outcomes. For example, outgroup prejudice is negatively associated with national identification but positively with national narcissism (Golec de Zavala, Cichocka, & Bilewicz, 2013).”

Importantly, studies conducted in the context of the COVID-19 pandemic suggested that collective narcissism might be associated with problematic attitudes and behaviours (e.g., Nowak et al., 2020 linked it to hoarding; Sternisko et al., 2020 linked it to conspiracy beliefs). Thus, it was important to test whether any desirable effects observed for national identification would be unique to it, or would be also observed for collective narcissism.

5.7 The authors surprisingly find that HDI was related to less public health support and note this might be driven by these African countries as outliers. They write, “In other words, citizens in countries with higher scores on the global Human Development Index also reported less support for two of our COVID-19 public health measures. However, we should note that our dataset includes data from very few African countries, many of which have relatively low HDI scores but seem thus far to have fared better in the pandemic than higher-HDI countries.”

We have moved this comment to a footnote since it was not a central analysis for our hypothesis or a finding we place a great deal of trust in for the reasons we mentioned. More importantly, we want to underscore that African countries did not drive the central effects in our paper (linking national identification with public health measures, as shown in Figure 3). The same general pattern was observed on almost every public health measure in every country we studied. Indeed, the countries with the strongest relationship between national identification and public health measures were: Denmark, China, Philippines, United States, Ukraine, Poland, Cuba, Republic of Korea, Japan, France, Latvia, Australia, Taiwan, Iraq, Hungary, and Russian Federation--none of which were in Africa. As such African countries were not outliers or even at the edge of the distribution and therefore could not drive any of the effects.

We hope you will agree that these changes have significantly improved the manuscript and hope it is now suitable for publication in *Nature Communications*.

Reviewers' comments:

Reviewer #2 (Remarks to the Author):

I am happy with the authors' responses, and to support publication of the article. I particularly note the results obtained from aggregate level data, which supports the individual level results.

Reviewer #3 (Remarks to the Author):

I find the revised paper to be very strong and that my concerns have been addressed. I recommend publication.

Reviewer #4 (Remarks to the Author):

I have read earlier versions of this manuscript and think it gets better every time. The authors have done an excellent job addressing reviewers' concerns. I continue to have quibbles here and there, but I fully recognize that the authors have undertaken a monumental gathering of data and their results are important. This work contributes not only to research on public health efforts and national identity, it also has practical implications for public health experts' responses in future pandemics (which there will be). I think the authors' emphasis on the need for nuance in these responses, given the impact of ideology and threat on national identity in calls for public health behaviors, is especially important. Overall I think the authors have an important piece of research here.

Reviewer #5 (Remarks to the Author):

I was the 5th reviewer in the previous round of reviews. I applaud the authors for constructively addressing most of my concerns. Although I am not fully convinced by the reliability and measurement invariance arguments, the authors justify their position satisfactorily and address it statistically through moderation. I especially congratulate the authors for running the suggested second study. I think it really lifts the contribution of this paper to a new level as it deals with actual behavior. Greatly done. This being said, I have some remaining, minor issues that should be easily addressed.

I thank the authors for providing new graphs with confidence intervals. I think these new graphs underline why the current Figure 2 should not be presented. Not only can it easily mislead an average reader to compare countries (despite non-representative samples), the figure also gives no information about the accuracy of these results. Eyeballing Figure S1, it seems as if the relationships are non-significant between national id and physical contact for 25 countries, between national id and hygiene for 14 countries, and between national id and policy support for 23 countries. This is essential information to the reader and needs to be conveyed. I, hence, suggest replacing the current figure with Figure S1, that can be adjusted to look more physically appealing (e.g., colors, change of axis ranges etc.).

Related to this, the authors should nuance some of their conclusions. For instance, they write in the abstract, "Respondents who identified more strongly with their nation consistently reported greater engagement in public health behaviors and support for public health policies". As the new figure reveals, this was less consistent than I initially thought.

I appreciate that the authors checked the manuscript for causal language. However, it still contains quite a bit of it, especially in central places such as the title and abstract. I reiterate that I think "predict" should be replaced with "is associated with" in the title. In the abstract the authors write "we investigated why people reported adopting public health behaviors". This implies a "because", i.e., a causal relation – something this paper cannot answer. In most places of the paper, the authors now

use correlational language. Why not use it here as well to most accurately convey information?

Small point:

I. 472-474 "The fact that nationalism is associated with large-scale behavior in real life provides ecologically valid evidence for our main hypothesis." Unless I misunderstood something, I would suggest sticking to the term national identity, as nationalism was not measured.

REVIEWERS' COMMENTS

Reviewer #2 (Remarks to the Author):

I am happy with the authors' responses, and to support publication of the article. I particularly note the results obtained from aggregate level data, which supports the individual level results.

We are grateful for this thoughtful response.

Reviewer #3 (Remarks to the Author):

I find the revised paper to be very strong and that my concerns have been addressed. I recommend publication.

We are grateful for this thoughtful response.

Reviewer #4 (Remarks to the Author):

I have read earlier versions of this manuscript and think it gets better every time. The authors have done an excellent job addressing reviewers' concerns. I continue to have quibbles here and there, but I fully recognize that the authors have undertaken a monumental gathering of data and their results are important. This work contributes not only to research on public health efforts and national identity, it also has practical implications for public health experts' responses in future pandemics (which there will be). I think the authors' emphasis on the need for nuance in these responses, given the impact of ideology and threat on national identity in calls for public health behaviors, is especially important. Overall I think the authors have an important piece of research here.

We are grateful for this thoughtful response.

Reviewer #5 (Remarks to the Author):

I was the 5th reviewer in the previous round of reviews. I applaud the authors for constructively addressing most of my concerns. Although I am not fully convinced by the reliability and measurement invariance arguments, the authors justify their position satisfactorily and address it statistically through moderation. I especially congratulate the authors for running the suggested second study. I think it really lifts the contribution of this paper to a new level as it deals with actual behavior. Greatly done. This being said, I have some remaining, minor issues that should be easily addressed.

We are grateful for this thoughtful response.

I thank the authors for providing new graphs with confidence intervals. I think these new graphs

underline why the current Figure 2 should not be presented. Not only can it easily mislead an average reader to compare countries (despite non-representative samples), the figure also gives no information about the accuracy of these results. Eyeballing Figure S1, it seems as if the relationships are non-significant between national id and physical contact for 25 countries, between national id and hygiene for 14 countries, and between national id and policy support for 23 countries. This is essential information to the reader and needs to be conveyed. I, hence, suggest replacing the current figure with Figure S1, that can be adjusted to look more physically appealing (e.g., colors, change of axis ranges etc.).

We respectively disagree with this suggestion. The figure we are using has been used in a recent Nature Communications piece as well as several recent papers in high profile journals (including Science). We believe readers are capable of analyzing the global pattern of means for all variables more efficiently with this figure (including seeing the cross-national heterogeneity the reviewer mentions). In our opinion, it is impossible to effectively communicate this information with all the CIs. As such, we have included those more detailed (by hard to read) figures in the supplement.

Related to this, the authors should nuance some of their conclusions. For instance, they write in the abstract, “Respondents who identified more strongly with their nation consistently reported greater engagement in public health behaviors and support for public health policies”. As the new figure reveals, this was less consistent than I initially thought.

We now note that this is an average effect across countries and the general pattern is highly consistent in terms of direction across virtually every country we measured and every DV we included. However, the pattern is not identical in each country which is why we’ve included figures to clearly display the heterogeneity (and the consistency of the general pattern).

I appreciate that the authors checked the manuscript for causal language. However, it still contains quite a bit of it, especially in central places such as the title and abstract. I reiterate that I think “predict” should be replaced with “is associated with” in the title. In the abstract the authors write “we investigated why people reported adopting public health behaviors”. This implies a “because”, i.e., a causal relation – something this paper cannot answer. In most places of the paper, the authors now use correlational language. Why not use it here as well to most accurately convey information?

We have sought to remove causal language throughout the paper. However, we have kept the term predict in the title because it is now technically correct since we added a second study in which measures of national identification effectively predicted subsequent aggregate behavior during the pandemic. (We further note that the term “predict” is also widely used in the social and behavioral sciences to describe clear associations and does not necessarily imply causation in many literatures).

Small point:

1. 472-474 “The fact that nationalism is associated with large-scale behavior in real life provides

ecologically valid evidence for our main hypothesis.” Unless I misunderstood something, I would suggest sticking to the term national identity, as nationalism was not measured.

Thanks for this note. We have removed the word nationalism as it connotes a slightly different construct.

We hope you will agree that these changes have significantly improved the manuscript and hope it is now suitable for publication in *Nature Communications*.